# EGFR-targeted affibody–polyIC polyplex kills EGFR-overexpressing cancer cells without activating the EGFR

Anne Pettikiriarachchi[1,2☉‡], Yelena Ugolev[3☉‡], Richard Birkinshaw[2], Ahmad Wardak[2], Maree C. Faux[1,2], Timothy E. Adams[4], Salim Joubran[3], Alexei Shir[3], Nufar Edinger[3], Maya Zigler[3], Alexander Levitzki[3], Antony W. Burgess[1,2,5*]

**1** Personal Oncology, Walter and Eliza Hall Institute of Medical Research, Parkville, Australia, **2** Structural Biology, Walter and Eliza Hall Institute of Medical Research, Parkville, Australia, **3** Unit of Cellular Signaling, Silberman Life Sciences Institute, The Hebrew University of Jerusalem, Jerusalem, Israel, **4** CSIRO Manufacturing, Clayton, Victoria, Australia, **5** Department of Surgery (RMH), University of Melbourne, Parkville, Australia

☉ These authors contributed equally to this work.
‡ These authors share first authorship for this work.
* tburgess@wehi.edu.au

## Abstract

The epidermal growth factor receptor (EGFR) is aberrantly activated in many human epithelial cancers. This report presents the preparation, purification, and the anti-cancer potency of an anti-EGFR affibody ($Z_{EGFR\ 1907}$)-polyethylenimine (PEI)-polyIC complex (PPEA-polyplex). Surface plasmon resonance analysis showed that the $Z_{EGFR\ 1907}$ affibody binds tightly to full-length sEGFR with an average equilibrium dissociation constant, $K_D$, value of 6.74 nM. As expected the PPEA-polyplex does not activate the EGFR kinase, but kills tumor cells expressing medium to high levels of EGFR. The PPEA-polyplex stimulates the release of chemotactic cytokines (e.g., GRO-α, IFN-γ–inducible protein-10) and promoted PBMC-mediated bystander killing of non-treated tumor cells. The PPEA-polyplex also inhibited the growth of human epidermoid vulval carcinoma (A431) xenografts growing in immunocompromised nude mice. Both the *in vitro* and *in vivo* results indicate that PPEA-polyplexes have the potential to inhibit the growth of tumors which over-express the EGFR, including colon and breast cancer cells.

## Introduction

The epidermal growth factor receptor (EGFR) was identified as a target for cancer therapy almost forty years ago [1]. Over 70% of malignant tumors have abnormal expression or activation of the EGFR, including brain [2], bladder [3], colorectal [4,5], head and neck [6], lung [7] and breast cancers [8]. Cancers can be driven by autocrine activation of the receptor [9,10], mutation [11] or amplification [12] of the EGFR gene. This perturbation of EGFR signaling homeostasis plays a critical role

**Data availability statement:** All relevant data are within the paper and its Supporting information files.

**Funding:** The author(s) received no specific funding for this work.

**Competing interests:** The authors declare that no competing interests exist.

in initiation, progression, growth and metastasis of most colorectal cancers (CRCs) [13,14]. Targeting cancers with EGFR receptor antagonists (e.g., antibodies or specific kinase inhibitors) has already helped millions of cancer patients world-wide [15]. In particular CRC patients without K-ras mutations and lung cancer patients with mutant EGFR have responded to EGFR antagonist treatment [16]. There are indications that combined, targeted therapies focused on the EGFR family might help many other cancer patients [17], including pancreatic [18], head and neck [19] and brain cancer patients [20].

Cancer immunotherapy has progressed, especially through the use of anti-CTL4A and anti-PD1/PDL1 in the treatment of metastatic melanoma and the emerging success of tumor targeting chimeric T-cells [21,22]. We have previously demonstrated that EGFR-targeted dsRNA polyinosine/polycytosine (polyIC) eradicates EGFR overexpressing tumors in experimental animals [23,24]. PolyIC has been known as an immune stimulating agent for dozens of years but was too toxic for systemic utilization. Yet, polyIC is used by local application to enhance the action of vaccines [25,26].

Tumor cells expressing high levels of EGFR can be killed with EGFR-homing chemical vectors loaded with polyIC [23,27]. In these previous studies [23,24,27] EGF was used to deliver the polyIC to the surface of the neoplastic cells and following endocytosis the delivery of the polyIC to the cytoplasm was facilitated by the polyethyleneimine-polyethylene-glycol (PEI-PEG) linker [24]. This PEI-PEG-EGF (PPE) polyIC complex (polyplex) induced tumor cell killing, both *in vitro* and *in vivo*. Moreover, we showed that following internalization, polyIC activates multiple cell-killing mechanisms and induces strong bystander effects, leading to killing of both targeted and non-targeted tumor cells, without harming the neighboring normal cells [23,24,27]. The resistance of normal cells, including cells which express low numbers of EGFR, is most likely due to the more robust nature of these cells as compared to tumor cells, which are under constant stress [28].

The use of EGF as the targeting moiety raises the possibility of activating tumor cells via the EGFR kinase and even causing reactions in normal progenitor cells or stem cells which express the EGFR. Anti-tumor responses have been obtained with agents which block the EGFR, e.g., cetuxumab, but it is not clear that a targeting reagent which does not activate the receptor kinase could deliver the polyIC into the cell. Affibodies are small molecules capable of specific targeting [29]; in particular affibodies such as $Z_{EGFR\ 1907}$ binds to the EGFR, but do not activate the EGFR kinase [29,30]. We decided to invetigate whether the affibodies could deliver the polyIC to cells despite not activating the EGFR kinase.

In 2007 Friedman and colleagues engineered a series of EGFR-binding affibodies [31]: $Z_{EGFR\ 1907}$ was determined to bind to the EGFR with high affinity, $K_D$ of 5.4 nM. Optimisation of the affibody scaffoldings led to development of a protein with improved properties [32] and further modifications resulted in an improved variant of $Z_{EGFR\ 1907}$ with increased thermal stability, $Z_{EGFR\ 1907'}$. The $Z_{EGFR\ 1907'}$ affibody binding to EGFR does not appear to induce receptor phosphorylation, but it is internalised by the cell [33,34]. In this report we engineer, characterise and evaluate the potency

of $Z_{EGFR\ 1907'}$-polyIC-polyplexes for targeting the EGFR. The $Z_{EGFR1907'}$ affibody was linked covalently to a LPEI-PEG diconjugate to form the LPEI-PEG- $Z_{EGFR1907'}$ triconjugate which targets the EGFR. The $Z_{EGFR\ 1907'}$-affibody-triconjugate (PPEA) was complexed with polyIC to form the anti-tumor $Z_{EGFR\ 1907'}$-affibody-polyIC complex (PPEA-polyplex). The potency of the anti-tumor activity of the PPEA-polyplex was investigated.

## Abbreviations and terminology

Abbreviations are appended to the full description of the chemicals when they first appear in the text, however it is important to note that the abbreviation PPEA-polyplex is a conjugate of polyethyleneglycol, polyethyleneimine, polyIC and the anti-EGFR affibody $Z_{EGFR1907'}$. When it is appropriate to emphasize that this polyplex contains polyIC, we also use the term PPEA-polyIC-polyplex for this conjugate. $Z_{EGFR\ 1907'}$ is used as an abbreviation for $Z_{EGFR\ 1907'}$ affibody.

The abbreviation sEGFR is used to describe extra-cellular fragments of the EGFR. We use two particular sEGFR fragments: residues 1–621 ($EGFR_{1–621}$) and residues 1–501 ($EGFR_{1–501}$).

We have analyzed cell lines with a wide range of EGFR expression; when cells have less than 5% of the level of receptors on DiFi cells they are described as low, cells with greater than 5 and less than 50% of the EGFR levels for Difi cells are described as moderate and cells with higher than 50% of the EGFR levels on Difi cells are described as high.

## Materials and methods

Plasmid DNA encoding the affibody $Z_{EGFR\ 1907'}$-Cys (an affibody which binds to the EGFR) was transformed into *Escherichia coli* BL21 (DE3). The transformed bacteria were grown overnight in 50 ml 2xYT media with 1% glucose and 30 µg/ml kanamycin at 37°C. This overnight starter culture was transferred into larger culture and allowed to grow to an $OD_{600nm}$ between 0.5 and 1.0. Protein expression was induced with 0.5 mM isopropyl-β-D-1-thiogalactopyranoside (IPTG) – overnight at 28°C. The bacterial cell pellet was stored at -80°C until required for the purification processes.

The bacterial cell pellet containing the anti-EGFR affibody was thawed and resuspended in 50 ml 20 mM Hepes pH 7.4, 500 mM NaCl, 10% glycerol, 10 mM Imidazole and 2 mM β-mercaptoethanol (Buffer A) and lysed using a homogeniser. The soluble proteins were recovered by centrifugation at 20,000 rpm for 20 min at 4°C. His-Tag EGFR affibody protein was purified using a Ni-NTA column. The column with the bound anti-EGFR affibody was washed with Buffer A for 15 column volumes (CV) and with Buffer A containing 30 mM Imidazole for 10 CV. Bound EGFR affibody was eluted using buffer A containing 500 mM Imidazole for 5 CV. The purity of the EGFR Affibody protein was assessed by SDS-PAGE analysis (Fig 1Ai). The eluted protein was then loaded onto a gel filtration chromatography column Superdex 75 16/60 (GE healthcare) equilibrated using 20 mM Hepes pH7.4, 500 mM NaCl, 10% Glycerol and 2 mM β-mercaptoethanol. 1 ml fractions from the peak spectrum were collected and the purity of the gel filtration purified samples were assessed by SDS-PAGE (Fig 1Bi and1Bii). Ellman's reagent (Thermo Scientific) was used to determine the presence of a sulphydryl group as per the manufacturer's protocol.

## SDS-polyacrylamide gel electrophoresis (SDS-PAGE) analysis of proteins

Samples of Ni-NTA–purified $Z_{EGFR-1907'}$ were diluted in SDS loading buffer with or without 1.43 M β-mercaptoethanol, heated to denature, and separated on NuPAGE 4–12% Bis-Tris gels (MES buffer; Invitrogen). Protein bands were stained with Coomassie Brilliant Blue R-250 (BioRad) (Fig 1Ai). The concentrated 1:1 Triconjugate $Z_{EGFR-1907'}$ affibody construct (see Synthesis of the LPEI-PEG-EGFR Affibody section below) was also analysed using SDS-PAGE. To visualise the sample bands, the gel was silver stained as follows: washed with 40 mM DTT, stained with 1 mg/ml $AgNO_3$, washed with de-ionized water, developed with 30 mg/ml $Na_2CO_3$ and 0.15% formaldehyde, reaction stopped with de-ionized water and gel fixed with 10% acetic acid (Fig 1C).

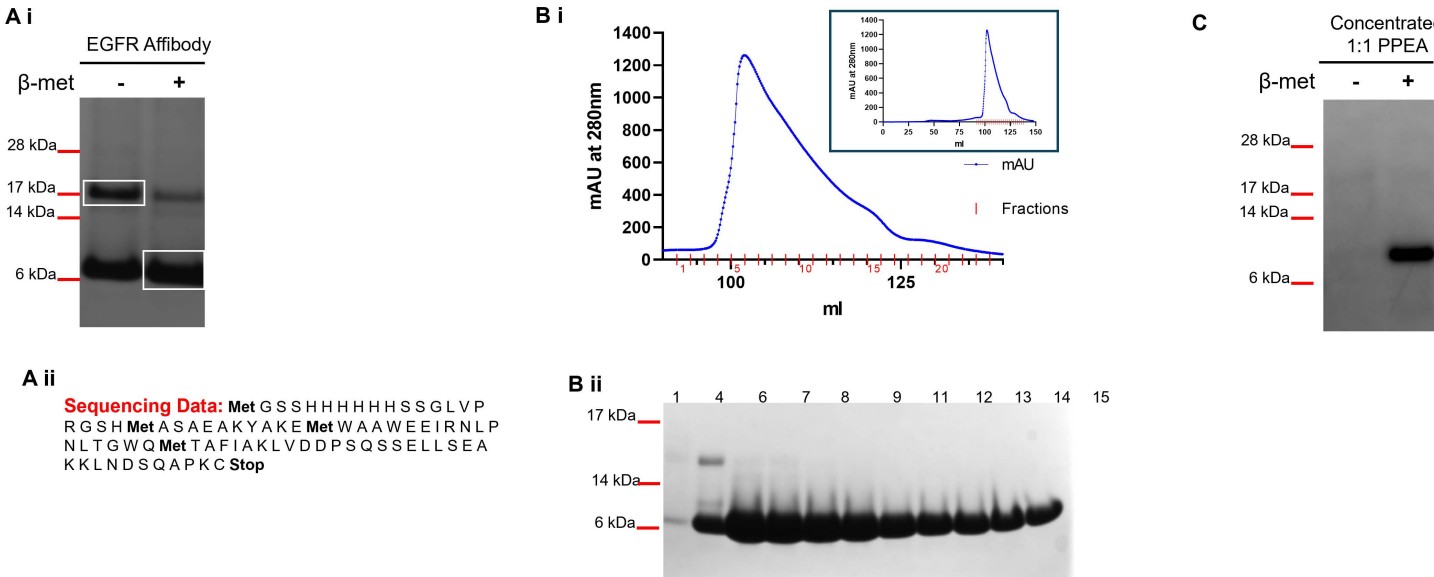

**Fig 1. Generation of the LPEI-PEG-Z$_{EGFR1907'}$. (A i)** SDS-PAGE analysis showing the purity of the Ni-NTA purified Z$_{EGFR 1907'}$ under pseudo native and denatured conditions. Pseudo native gel bands (boxed) were extracted and analysed by mass spectrometry. Mass spectrometry data confirmed that both bands correspond to one species, **(A ii)** DNA sequencing of the plasmid revealed the above sequence and this was also confirmed by mass spectrometry analysis **(B i)** Elution profile of the size exclusion gel filtration chromatography of Z$_{EGFR 1907'}$ affibody purified via His-Tag Ni-NTA purification. Superdex 75 1660 column equilibrated with 20 mM Hepes pH 7.4, 500 mM NaCl, 10% Glycerol and 2 mM β mercaptoethanol and 1 ml fractions were collected. Insert shows the complete elution profile. **(B ii)** 20 µl of selected sample fractions from the purification were denatured, run on an SDS-PAGE gel and stained with Coomassie blue. **(C)** Silver stained SDS-PAGE gel of the concentrated 1:1 triconjugate Z$_{EGFR 1907'}$ (PPEA) under pseudo native and denatured conditions. Pseudo native and denatured conditions are indicated by -/+ β-mercaptoethanol.

## Mass spectrometry analysis

Protein bands responding to the Z$_{EGFR 1907'}$-affibody were extracted from a Coomassie stained SDS-PAGE gel and subjected to manual in-gel tryptic digestion. Briefly, the gel spots were de-stained and dehydrated with acetonitrile before being submitted to reduction with 50 mM TCEP solution (50 µL of 0.5 M TCEP / 125 µL of 200 mM TEAB / 325 µL deionized water) and alkylation with 100 mM Iodoacetamide solution in 50 mM TEAB buffer. Trypsin solution was prepared by adding 50 µL of 50 mM TEAB to each trypsin single vial (1 µg trypsin) and rehydrating dried gel plug/bands with approximately 10 µL trypsin working solution for 60 min at 4°C. Excess trypsin solution was removed with a pipette before adding 35 µL of 50 mM TEAB to the sample vial to ensure proper hydration during digestion at elevated temperatures. Digestion was performed at 37°C overnight. The digestion was stopped by adding 5 µL of 10% formic acid before collecting the supernatant. The generated tryptic peptides were concentrated to ~20 µL by centrifugal lyophilization for LC MS/MS analysis for peptide mapping mass spectrometry on LC/MS/MS.

Mass spectrometry was used to confirm protein purity, molecular mass and the protein sequence (Fig 1Aii). A spectrometer (Thermo Scientific) with a nanoESI interface in conjunction with an Ultimate 3000 RSLC nano HPLC (Dionex Ultimate 3000) was used. The LC system was equipped with an Acclaim Pepmap nanotrap column (Dionex-C18, 100 Å, 75 µm x 2 cm) and an Acclaim Pepmap RSLC analytical column (Dionex-C18, 100 Å, 75 µm x 15 cm). The tryptic peptides were injected into the enrichment column at an isocratic flow of 5 µL/min of 3% v/v CH3CN containing 0.1% v/v formic acid for 5 min applied before the enrichment column was switched in-line with the analytical column. The eluents were 0.1% v/v formic acid (solvent A) and 100% v/v CH3CN in 0.l% v/v formic acid (solvent B). The flow gradient was (i)

0–5 min at 3% B, (ii) 5–6 min, 3–6% B (iii) 6–18 min, 6–10% B (iv) 18–38 min, 10–30% B (v) 38–40 min, 30–45% B (vi) 40–42 min, 45–80% B (vii) 42–45 min at 80% B (viii) 45–46 min, 80–3% B and (ix) 46–53 min at 3% B. The LTQ Orbitrap Elite spectrometer was operated in the data-dependent mode with nano ESI spray voltage of 2.0 kV, capillary temperature of 250°C and S-lens RF value of 55%. All spectra were acquired in positive mode with full scan MS spectra scanning from m/z 300–1650 in the FT mode at 240,000 resolutions after accumulating to a target value of $1.0 \times 10^6$. Lock mass of 445.120025 was used.

The top 20 most intense precursors were subjected to collision-induced dissociation (CID) with normalized collision energy of 30 and activation. The generated MGF files were uploaded to the Bio21 MSPF Pipeline (Bio21 Mass Spectrometry and Proteomics Facility Pipeline, http://proteomics.bio21.unimelb.edu.au/msile) and then searched using the MASCOT version 2.4.01 for algorithm against the murine Uniprot fasta database that included the added affibody fasta sequences (84562 sequences in total). The search parameters consisted of carbamidomethyl of cysteine as a fixed modification, NH2-terminal acetylation and oxidation of methionine as variable modifications. Up to three missed tryptic cleavage sites were allowed (trypsin/P) with a peptide mass tolerance of 20 ppm, a fragment mass tolerance of 0.8 Da, automatic decoy searches enabled with low decoy rate (< 5%), a significance threshold $p < 0.05$ and a Mascot score >30.

## Biosensor analyses

SPR binding assays (Biacore S200, Cytiva) were performed to measure interactions of $Z_{EGFR\,1907}$, and Cetuximab with sEGFR$_{1–501}$ and sEGFR$_{1–621}$. Proteins were immobilized on CM5 chips by amine coupling (pH 5.0) to ~1500 RU (sEGFR$_{1–501}$) and ~1700 RU (sEGFR$_{1–621}$). An initial pH scouting experiment was performed using 10 mM Acetate with pH 4.0, 4.5, 5.0, 5.5 and 6.0 for sEGFR proteins, sEGFR$_{1–501}$ and sEGFR$_{1–621}$. Samples were injected over the flow cell surface at 10 µL/min for 12 min. EGFR proteins were diluted in SPR buffer, 10 mM Hepes pH 7.4, 150 mM NaCl, 3 mM EDTA and 0.005%(v/v) Tween 20. EGFR proteins, sEGFR$_{1–501}$ and EGFR$_{1–621}$, were immobilized at 1500 RU and 1700 RU (resonance units) respectively to Series S Sensor Chip CM5 (Cytiva) by amine coupling at pH 5.0 (Table S1), according to the manufacturer's instructions.

The conditioning cycle consisted of 0.1% SDS for 60 sec, 100 mM HCl for 60 sec, 10 mM NaOH for 60 sec and 50 mM glycine pH 2.5 for 60 sec. For binding kinetic studies, proteins were injected at 30 µL/min for 300 sec association over the sEGFR surfaces. $Z_{EGFR\,1907}$, and Cetuximab concentrations ranged from 0 to 250 nM. This was followed by 800 sec dissociation phase. After each injection, the sensor surface was regenerated with SPR buffer supplemented with 50 mM glycine pH 2.5 for 120 sec between each cycle before repeating the sample injection. The dissociation equilibrium constant ($K_D$) was calculated using Biacore S200 Evaluation Software 1.1 (Cytiva). The curves were fitted using a local 1:1 binding model and a global heterogenous binding model.

The binding sites of the $Z_{EGFR\,1907}$, and Cetuximab to sEGFR$_{1–621}$ were explored. For A-B-A competitive binding assays, various concentrations and combinations of competitor sample solutions (B from A-B-A assay) and flanking competitor sample solutions (A from A-B-A assay) were tested. The assay consisted of 200 sec of flanking solution A, followed by 180 sec of sample solution B and further 60 sec of flanking solution A. Protein samples were pumped over the cell at a rate of 30 µL/min. The sensorgrams generated were subtracted from reference flow cell values. After each condition the flow cell was regenerated with SPR buffer supplemented with 50 mM glycine pH 2.5. A flow chart, summarizing the steps in the synthesis of the affibody-polyplex is presented in the supplementary information (Supplementary Figure S1). The details of each phase of the synthesis are documented in the sections below.

## Synthesis of LPEI-PEG-OPSS thiol reactive co-polymer

Linear polyethyleneimine (LPEI) precursor, poly(2-ethyl-2-oxazoline), was synthesized following the methods published by Brissault and colleagues [35,36].

In brief, methyl p-toluenesulfonate (74 mg, 397 mmol) dissolved in 60 ml acetonitrile was added to 8 ml (79 mmol) of 2- ethyl-2-oxazoline. The reaction mixture was stirred under reflux for 2 hrs, during which time a precipitate formed. The crude product was dissolved in 100 ml of methylene chloride and precipitated in 400 ml of diethyl ether yielding 5.5 g of poly(2-ethyl-2-oxazoline) as a yellow powder. Synthesis of the LPEI (free base form) was prepared as described previously [36,37]. In brief, 5.5 g (0.11 mmol) of poly(2-thyl-2-oxazoline) was hydrolyzed with 68.8 ml of concentrated HCl (37%). The resulting air-dried LPEI hydrochloride salt was dissolved in 100 ml of water to yield 4 g of LPEI salt. This was made alkaline by adding 100 ml of 3 M NaOH. The lyophilised solid weighted ~1.25 g. LPEI-PEG$_{2k}$-OPSS diconjugates (1:1 and 1:3) were synthesized as previously described [37]. 174 mg (8 μmol) of LPEI in 2.7 ml of absolute EtOH was mixed with 79 mg of OPSS-PEG2k-CONHS (39.5 μmol) in 500 μl of anhydrous DMSO. The co-polymers conjugated with different molar ratios of PEG to LPEI were separated by cation exchange chromatography using a HR10/10 column packed with MacroPrep High S resin (BioRed).

Upon agitation for 3 hrs, 2 ml of 20 mM Hepes pH 7.4 were added to the reaction mixture and after few min, once the viscosity increased, the pH was adjusted to 7.4 with 1 M HCl. The column was equilibrated with 20 mM Hepes pH 7.4 buffer (buffer A). Separation was performed under a 20 mM Hepes pH 7.4; 3 M NaCl (buffer B) gradient. Sample was loaded at 1 ml/min for 7 min and for further 60 min with 0% buffer B. This was followed by isocratic elution at 20% buffer B until 76 min, followed by gradient elution up to 45% buffer B at 87 min and continuing at 45% buffer B until 110 min. Then, isocratic elution at 50% buffer B until 120 min was followed by gradual increase in gradient up to 100% buffer B until 180 min. Isocratic elution continued at 100% buffer B until 215 min, followed by a gradual decrease in gradient to 0% buffer B by 218 min. 2 ml fractions were collected between 0% − 50% Buffer A and 1 ml fractions were collected during the elution step (50% −100% buffer B). Sample separations were monitored at 220 nm, 280 nm and 343 nm, and all fractions from the three peaks were stored -20°C.

Fractions from the 1:1 PEG to LPEI conjugation, which eluted during the 50% to 100% buffer B, were combined and desalted against 20 mM Hepes buffer pH 7.4 using a 20 ml HiTrap desalting column (4 x 5 ml Sephadex G25 columns). The purest fractions containing LPEI-free LPEI-PEG-OPSS were pooled and stored in the dark at -20°C. A copper assay [38] was used to determine the concentration of di-conjugate 1:1, with molar ration of LPEI to PEG ~ 1:1. In brief, copper assay consisted of incubating copolymers (initially dissolved at 2x concentration with 100% methanol and then diluted to 1x with de-ionized water) with CuSO$_4$ (23 mg of CuSO$_4$.5H$_2$O in 100 ml of 0.1 M acetate buffer pH 5.4) for 20 min and measuring their absorbance at 285 nm.

## Synthesis of the LPEI-PEG-EGFR affibody

The 1:1 diconjugate was mixed with EGFR affibody as described by Joubran *et al.* [37]. The reaction mixture was vortexed at ~800 rpm for 24 hrs and stored at -20°C. Later, the reaction mixture was thawed and purified using cation exchange chromatography using a HR10/10 filled with MacroPrep High S resin (BioRad). 3 step gradient elution from 20 mM Hepes pH 7.4 reaching to 20 mM Hepes pH 7.4 with 3 M NaCl was used. Sample was loaded at 1 ml/min for 6 min and for further 10 min with 0% buffer B. Then isocratic elution with 33% buffer B until 15 min was followed by gradient elution reaching 66% buffer B at 30 min, isocratic elution from 30 min to 37 min, and gradient elution reaching 100% buffer B at 55 min. Finally, there was a gradual decrease down to 33% buffer B at 60 min. 1 ml fractions were collected between 0% − 100% Buffer B. Sample separation was monitored at 213 nm, 280 nm and 343 nm, and all fractions from the three peaks were stored at -20°C. The 1:1 triconjugate fractions eluting between 25 min and 32 min (i.e., the last peak eluted by the gradient) were combined and desalted using the 4x5 ml Sephadex G25 (two runs) against PBS. Desalted fractions were combined, and 1 ml aliquots were frozen at -80°C. The concentration of the 1:1 triconjugate was determined using the copper assay. The triconjugate fractions were pooled, concentrated and buffer exchanged to 20 mM Hepes pH 7.4, 5% glucose with RNase free de-ionized water (HBG) using a 3K MWCO Amicon Ultra15 Centrifugal filter unit. The presence of the affibody (~8 kDa) was confirmed under denaturing/reducing conditions (Fig 1C).

## EGFR affibody polyIC-polyplex formation

The 1:1 Triconjugate affibody was complexed with poly (I:C) (Tocris, UK) in HBG buffer at a Nitrogen to Phosphate (N/P) ratio of 6 (where N = nitrogen from LPEI and P = phosphate from polyIC [37]). This ratio corresponds to a LPEI/polyIC weight ratio of 0.78 [39]. Both the 1:1 Triconjugate Affibody and polyIC were diluted up to twice the final volume using HBG buffer. Fresh polyplexes were made prior to the experiment by transferring equal volumes of 1:1 Triconjugate to the diluted polyIC and mixing by pipetting. The polyplexes were incubated at room temperature for 30 mins. The final 1:1 Triconjugate Affibody-polyIC-polyplex ($Z_{EGFR\ 1907'}$-polyIC-polyplex) was determined to be 100 µg/ml using the copper assay [36].

## Polyplex size measurements

To determine the influence polyIC concentration has on the polyplexes formed, the particle size and its distribution were measured at 25°C by dynamic light scattering using a Zetasizer µV (Malvern Instruments). The instrument was fixed with an 830 nm laser wavelength and the optic arrangement angle at which the measurements were performed was 90°. Each polyplex sample was measured in triplicate.

## Cell culture

EGFR-overexpressing A431 human epidermoid vulval carcinoma [40], SK-BR-3 [41], U87MG human glioma and its EGFR-overexpressing subline U87MGwtEGFR [42] cell lines were grown in Dulbecco's modified Eagle's medium (DMEM). U87MGwtEGFR cells were supplemented with geneticin sulfate (G418, Gibco, UK) at a final geneticin concentration of 500 µg/ml. MCF7, MDA-MB-231, MDA-MB-468 and BT-474 [43] cells were grown in Roswell Park Memorial Institute (RPMI). U138MG cells were grown in Eagle's Minimum Essential Medium (MEM) supplemented with 2 mM L-glutamine, non-essential amino acids and 0.1 g/L sodium pyruvate. All cell culture media were supplemented with 10% fetal bovine serum (FBS),100 U/mL penicillin, and 100 µg/mL streptomycin and all cells were grown at 37 ºC in an incubator with humidified air equilibrated with 5% $CO_2$.

CRC cell lines: Difi, SW620, Lim2537, SW480, Lim2099, HCA7, Colo201, Colo320, HCT116, IS1, SW48, MC38, SW1116 and CX1 were obtained from colleagues either at WEHI or the Olivia Newton John Cancer Research Institute (ONJCRI). Other cell lines such as A431 (human squamous), were obtained from colleagues at WEHI, BT-20 and MDA-MB-468 were purchased from ATCC. All cell lines were grown in their recommended media (DMEM from Gibco, RPMI 1640 from Sigma and EMEM from ATCC) supplemented with 10%FCS, -/+Adds (1.08% thioglycerol, 50 mg/ml hydrocortisone, 100 U/ml insulin), -/+G418. All cell lines were grown to approximately 75% confluency before passaging. All cell lines except MC38 (mouse colon), BT-20 (human breast) and MDA-MB-468 (human breast) cell lines were maintained at 37°C and 10% $CO_2$. MC38, BT-20 and MDA-MB-468 cell lines were grown at 37°C and 5% $CO_2$. The original sources, current locations of these CRC cell lines and the characteristics of each cell line are documented in the Supplementary material of Wang *et al.* [44].

## Cell viability assay

Briefly, the optimal seeding cell density for each cell line was determined in triplicate in a 96 well plate: Difi 3000 cells per 90ul, BT20 2000 cells per 90ul, MBA-MB-468-1500 cells per 90ul, Lim1215 2500 cells per 90ul and SW620 2000 cells per 90ul.

Using optimal seeding densities, cell viability assays were set up in triplicates in 96 well plates. After 24hr, cells were treated with the varying concentrations PPEA-polyplexes and a cytotoxic drug such as 2.5µM WEHI 7326 [45] or 1µM Bortezomib (Cell Signaling Technology), for total cell death control. Untreated cell viability was determined by adding an equivalent volume of the HBG buffer. Cells were treated with drugs for either 2hr, 72 hr or 96 hr. Cell survival was quantitated using CellTiter96Aqueous One Solution Cell Proliferation assay or CellTiter-Glo 2.0 Cell Viability assay (Promega,

 

Madison) and luminescence plate reader (EnVision 2100 Multilabel reader) with Wallac EnVision Manager 1.12 software. Total absorbance for treated wells was calculated by subtracting the mean background (non-treated). Percentage of treated wells was calculated dividing total absorbance with mean values of non-treated minus total cell death. Mean percentage of treated wells was calculated by taking the average of percentage of the 3 treated cells and its SD. $EC_{50}$ was determined to have 50% of cell viability.

### Flow cytometric EGF receptor quantitation

For flow cytometric analysis $0.5 \times 10^6$ cells in 50 µl were seeded into U-bottom 96 well plates and the cell suspension was treated with 2.5 µg/ml mouse anti-human EGFR monoclonal antibody 528 (WEHI) and 0.4 µg/ml F(ab')2-Goat anti-Mouse IgG (H+L) PE (eBioscience) or 1.5 µl of PE anti-human EGFR antibody (BioLegend). Cells were incubated on ice for 1hr for anti-mouse 528 and 30 mins for PE antibody or 45 min for PE anti-human EGFR antibody. All samples except control cell preparations were stained with either 5 µg/ml Propidium Iodide (PI) (Thermofisher) or 0.1 µg/ml DAPI (Thermofisher). Cells expressing fluorescent protein and/or cell surface markers were analysed using CytoFLEX (Beckmann Coulter) or Fortessa1 (BD Biosciences) flow cytometers. Each analysis consisted of at least 10,000 live cells (events). Spectral bleed-through between the PI channel and PE channel were corrected by compensation. A431 and/or Difi cells were run as positive controls every time. FlowJo 10 software (BD Biosciences) was used for flow cytometric data analyses. Cells were gated in order: live cells, single cells and PI or DAPI negative cells. The final sorted population (PI negative cells) was analysed for presence of GFP using the 525 nm channel and PE using the 585 nm channel, and calculations on fluorescence intensity performed. The median values for the 585 nm population in quadrant 3 were obtained by subtracting the median PE value from median 528 nm, i.e.,1°AB+PE. The level of EGFR expression was expressed as a percentage of A431 or Difi EGFR expression. The error bars represent the standard deviation of the difference between sample medians ($\sigma_d$) [46]. $\sigma_d$ was calculated using the formula = sqrt ($\sigma_1^2 / n_1 + \sigma_2^2 / n_2$) where $\sigma_1$ and $\sigma_2$ refers to the standard deviations from sample 1 (528 1°AB+PE) and sample 2 (PE). $n_1$ and $n_2$ refers to the number of cells measured in each of these samples.

### PBMC extraction

Peripheral blood mononuclear cells (PBMC) from healthy human donors were separated on Ficoll Plaque (Pharmacia). Briefly, cells were diluted in RPMI 1640 (x4) and 40 mL of diluted cell suspension was layered over 13 mL of Ficoll Plaque in 50 mL conical tubes. Following centrifugation at 400g for 30 mins at 20 ºC in a swinging bucket rotor without brakes, the mononuclear cell layer was transferred to a new 50 mL conical tube, washed twice with 50 ml RPMI 1640 medium, resuspended at a density of $4 \times 10^6$ cells/ml and cultured in this RPMI 1640 medium supplemented with 10% fetal calf serum, 100 units/ml penicillin and 100 µg/ml streptomycin.

### Chemotaxis assay

Chemotaxis was performed as described previously [27], except that chemotaxis plates were from Corning (HTS Transwell-96 well plate, 5 µm polycarbonate membrane, catalogue no.3388).

### Cytokine measurements by ELISA

Supernatants from polyIC-treated cells were harvested and assayed for IP-10 and Gro-α using commercial ELISA kits (PeproTech) according to the manufacturer's instructions.

### Cell treatment and immunoblotting

MDA-MB-468 cells were seeded in 6-well plates (600,000 cells/well), allowed to grow for 24 hrs in compete medium and starved for 24 hrs in serum-free medium. PPEA was diluted in HBS (20 mM Hepes pH 7.4, 150 mM NaCl) and cells were

treated in duplicates with 120 ng PPEA for different periods of time. As a positive control for EGFR kinase activity, cells were treated with 10 ng EGF for 5 min. Cells were lysed with boiling sample buffer (10% glycerol, 50 mmol/L Tris-HCl pH 6.8, 3% SDS, and 5% 2-mercaptoethanol). Protein determination was performed using the bound Coomassie Blue method, as described previously [47]. Lysates containing 60 µg of protein were resolved on 10% SDS-PAGE gel. Western blot analysis was conducted as described previously [48]. The following antibodies were used: Antibody specific for EGFR was purchased from Santa Cruz Biotechnology (catalogue no.sc-03, dilution 1:1000); Antibody recognizing phosphory-lated EGFR (Tyr1068) was from Cell Signaling Technology (catalogue no. 2234S, dilution 1:1000); Anti-GAPDH was from Cell Signaling Technology (catalogue no. 2118, dilution 1:1000).

### Treatment of subcutaneous xenografts

Female nude mice (NUDE-HSD: Athymic Nude-NU mice) aged 3–4 weeks were obtained from Harlan, Rehovot. All animal experiments were performed according to the Hebrew University Ethical Committee regulations. Humane endpoints required by HUJI Ethical committee to reduce risk of pain to the animals:

1. Tumor length reaches 15 mm.

2. 10% decrease in weight between 2 consecutive weighings or 20% between 1st and last weighings.

3. Slow movement, apathy

A total of 40 animals were used in this experiment. Suffering of the animals was reduced to the minimum and animals were checked at least 3 times/week until the tumor diameter reached 10 mm; after which the animals were checked daily. Where possible, mice were sacrificed before reaching one of the formal humane endpoints. The total duration of the experiment was 21 days.

For A431 xenografts, two million A431 cells resuspended in 0.2 ml of PBS were injected subcutaneously into the right flank of immune compromised female athymic nude mice. The volume of growing tumors was calculated as follows: $V = LW^2/2$ (L = length and W = width). When the tumors reached an average volume of 137 mm³, mice were randomly divided into 5 groups (n = 7 or 8 per group), and the treatment was initiated. PolyIC formulated with PPEA in HBG buffer at w/w ratio of 0.78, which correlates with N/P ratio (molar ratio of nitrogen in PEI to phosphate in polyIC) of 6 as described previously [23,27], was injected. The polyplexes were administered via intravenous injection at doses of 0.75 mg/kg or 0.1 mg/kg every 24 hrs, 6 days per week, for a total of 9 injections over a 10-day period. Tumor volumes were measured on days 3, 6 and 10 after starting the intravenous injections.

### Ethics statement

All animal procedures were approved by the [Hebrew University Ethical Committee / relevant authority] under protocol number NS-13-13709-5 and conducted in accordance with institutional and national guidelines for the care and use of laboratory animals. De-identified human PBMC were obtained from the Israel National Blood Bank.

## Results

### Engineering and characterisation of triconjugate affibody-polyIC-polyplexes

We engineered an affibody-PEG-polyIC-polyplex to target the EGFR and deliver polyIC to the cytoplasm of cancer cells. An analogue of the anti-EGFR affibody [31], $Z_{EGFR\ 1907}$-Cys was expressed in *E.coli* BL21 (DE3) and purified on a Ni-NTA column from lysates as described in the methods section. SDS-PAGE analysis of the sample detected two bands with apparent molecular weights ~8 kDa and 16 kDa under pseudo native conditions (Fig 1Ai; see Materials and methods). These bands were analysed by mass spectrometry and the resulting sequence for the $Z_{EGFR\ 1907}$-Cys affibody is shown

in Fig 1Aii. The 8 kDa and 16 kDa bands correspond to the monomer and dimer of the $Z_{EGFR\ 1907'}$-affibody. The monomeric form of the $Z_{EGFR\ 1907'}$-Cys affibody was purified by gel filtration (Fig 1Bi). All subsequent experiments were performed using the purified with the $Z_{EGFR\ 1907'}$-Cys affibody monomer.

## Binding of the $Z_{EGFR\ 1907'}$-Cys affibody to the extracellular domain of the EGFR

Surface plasmon resonance analysis was used to measure the binding of the monomeric $Z_{EGFR\ 1907'}$-Cys affibody to two forms of the extracellular domain of EGFR: sEGFR$_{1-501}$ (a truncated form of the extracellular domain) [49] and sEGFR$_{1-621}$ (the full length extracellular domain) [50–52] (Fig 2A). Our binding kinetics data indicates that $Z_{EGFR\ 1907'}$-Cys affibody binds more tightly to EGFR$_{1-621}$ than EGFR$_{1-501}$ and has an average $K_D$ value of 6.74 ± 0.91 nM SD (Supplementary Figure S3A and Table 1). $Z_{EGFR\ 1907}$ has been reported previously to have a $K_D$ value of 5.4 nM using a similar method [31], however, different EGFR analogues were immobilized onto the biosensor. The strong affinity of the $Z_{EGFR\ 1907'}$-Cys affibody to EGFR$_{1-621}$ is consistent with having a more accessible groove created by folding of domain I and III in an untethered conformation by the full-length extracellular domain sEGFR$_{1-621}$ [53]. However, fitting the data to a global model suggests that the $Z_{EGFR\ 1907'}$-Cys affibody binds to two sites on the sEGFR$_{1-621}$ receptor with two different binding affinities (Table S2 and Supplementary Figures S2 and S3B). This was not evident when affibody bound to sEGFR$_{1-501}$. This finding supports the notion that sEGFR$_{1-621}$ adopts two conformations, tethered and untethered [54–56].

## Comparison of the binding of the ZEGFR 1907' and Cetuximab to sEGFR

For comparison purposes we studied the binding kinetics of Cetuximab [57,58] binding to the EGFR extracellular domain. Binding kinetic curves for Cetuximab showed that once bound to either sEGFR proteins, Cetuximab is slow in coming off the receptor as indicated by the flat dissociation curves (Supplementary Figure S3B). Comparison between $Z_{EGFR\ 1907'}$ and Cetuximab shows Cetuximab does not come off the EGFR sensor chips as easily as the $Z_{EGFR\ 1907'}$ (Supplementary Figure S3A). The tight binding of Cetuximab to the EGFR sensor reduces the ability of the chip to reach equilibrium binding, so the $K_D$ value determined in this study is not accurate. Our study determined Cetuximab to have an average $K_D$ value of 5.3 pM ± 9.9 SD for EGFR$_{1-621}$ which is higher affinity than the reported $K_D$ value of ~1 nM for Cetuximab binding to EGFR [59]. There was a significant difference between $K_{on}$ and $K_{off}$ values calculated for both forms of the sEGFRs under local 1:1 binding model fits (Table S3). It is possible that upon binding of Cetuximab to the EGFR surface, the conformation of the EGFR changes from a low affinity receptor to a high affinity binding state (Table 1). X-ray crystallographic and binding studies have shown that Cetuximab binds exclusively to domain III of soluble extracellular region of the EGFR (sEGFR) [53,60].

A competition screen between $Z_{EGFR\ 1907'}$ affibody and Cetuximab was measured using an A-B-A assay format where a flanking solution was injected before and after the sample. We used this assay to investigate the binding site of the $Z_{EGFR\ 1907'}$ affibody to the EGFR. As shown in Figs 2B, 2C, and S2B binding of Cetuximab blocks binding of $Z_{EGFR\ 1907'}$ affibody to sEGFR$_{1-621}$. The slight dip in response units upon injection of the second component (B: $Z_{EGFR\ 1907'}$ affibody) to Cetuximab suggests occurrence of some non-specific binding as Cetuximab is not expected to come off within 400 sec. Also, the relatively fast off rate for $Z_{EGFR\ 1907'}$ affibody allows Cetuximab to bind to newly unbound sEGFR$_{1-621}$ sites, however when Cetuximab is mixed with $Z_{EGFR\ 1907'}$ affibody in excess, this does not occur, indicating direct competition between the EGFR binders. Together these data show that the $Z_{EGFR\ 1907'}$ affibody shares the Cetuximab binding site (i.e., EGFR-domain III).

## EGFR expression and cell surface availability in colorectal and breast cancer cell lines

EGFR is overexpressed in majority of (50–80%) of CRCs(61) and these elevated levels of expression have been linked with tumor aggression and poor survival [14,61]. Cancer cells overexpressing EGFR appear to be ideal candidates for new targeted therapeutics such as the affibody polyplexes but was not clear that all the EGFR would be on the cell surface and accessible to the polyplexes. Consequently, we compared the published proteomic data from our laboratory, i.e., total EGFR levels [44] with the cell surface EGFR levels measured by flow cytometry for a range of CRC cell lines and

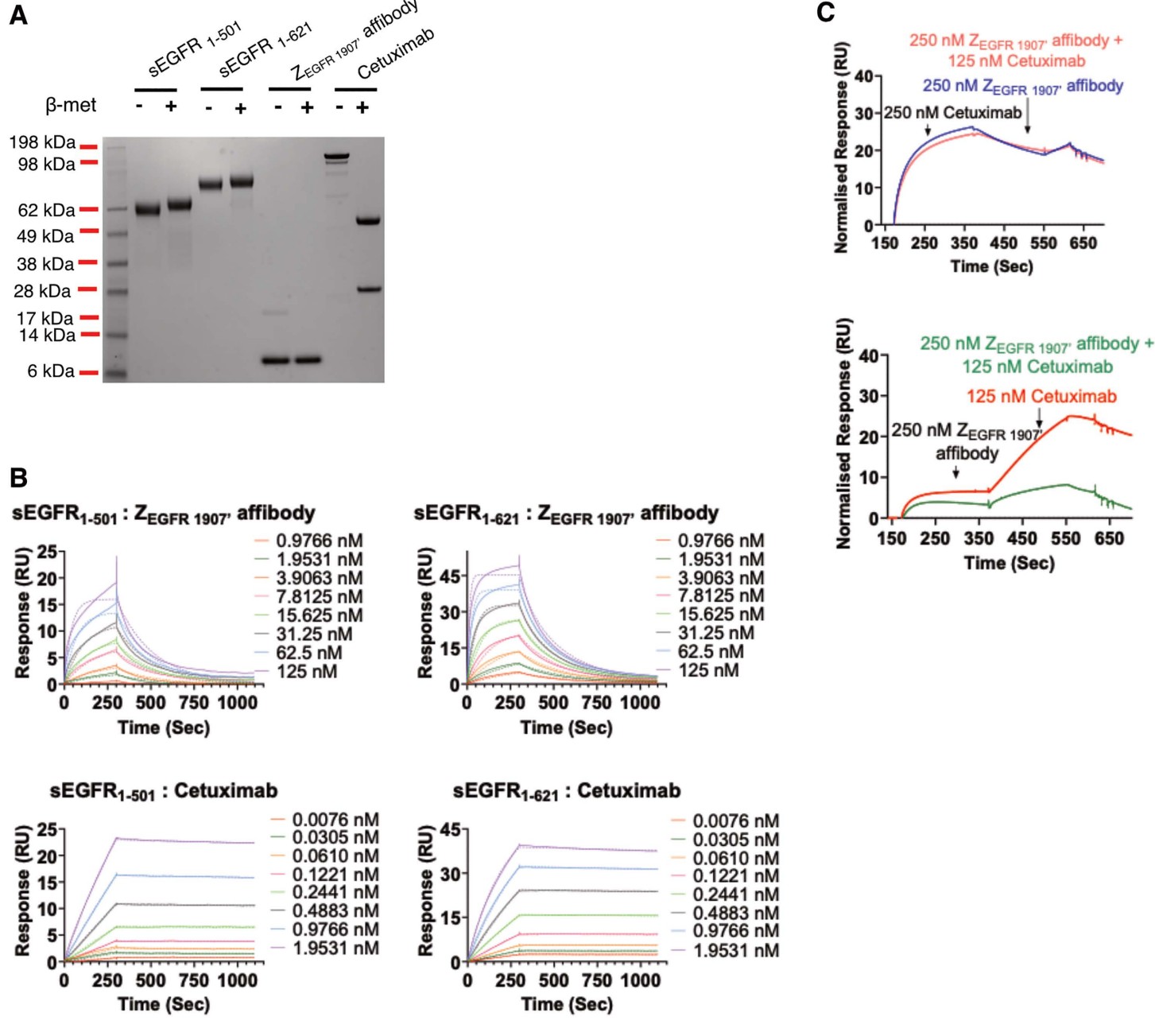

**Fig 2. Biacore analysis of Z_EGFR 1907' binding to EGFR proteins.** SDS-PAGE analysis showing the purity of the proteins used for the Biacore analyses. Expected molecular weights: sEGFR₁₋₅₀₁ 56kDA, sEGFR₁₋₆₂₁ 69kDa, Z_EGFR 1907' 11kDa and Cetuximab 152kDa. **(B)** Binding kinetics for the Z_EGFR 1907' and Cetuximab interacting with sEGFR₁₋₅₀₁ and sEGFR₁₋₆₂₁, used to determine K_d values using Biacore S200. **(C)** Competitive binding kinetics of Z_EGFR 1907' and Cetuximab detected under various conditions shows Cetuximab and Z_EGFR 1907' appear to bind to the same EGFR binding site. Solid and dotted line graphs represent raw and fitted data respectively.

some breast cancer cell lines. Cell lines with increased EGFR expression from the proteomic analyses also had more cell surface EGFR (Fig 3A, Spearman correlation coefficient ($r_s$) 0.75, p<0.001). For example, Difi cells have the highest levels of EGFR expressed on the cell surface (Fig 3A). A small number of CRC cells, e.g., SW620 and Colo201 and SW480

**Table 1. Binding affinities of Cetuximab and Z$_{EGFR\ 1907'}$ affibody to the EGFR-extracellular domain (ECD).**

| Sample | EGF-ECD fragment | k$_{on}$(1/Ms) x 10$^{-3}$ | k$_{off}$(1/s) x 10$^3$ | K$_D$(nM) | Model |
|---|---|---|---|---|---|
| Z$_{EGFR\ 1907'}$ affibody | sEGFR$_{1-501}$ | 240 ± 70 | 5.46 ± 0.6 | 23.3 ± 4.75 | 1:1 Binding |
| | sEGFR$_{1-621}$ | 1221 ± 380 | 8.0 ± 1.4 | 6.74 ± 0.91 | 1:1 Binding |
| Cetuximab | sEGFR$_{1-501}$ | 950 ± 98 | 0.034 ± 0.34 | 0.037 ± 0.006 | 1:1 Binding |
| | sEGFR$_{1-621}$ | 2600 ± 300 | 0.012 ± 0.02 | 0.005 ± 0.009 | 1:1 Binding |

Note: The values shown on the table represent the average of 3 individual experiments with ± values representing the standard deviation (SD).

showed little to no EGFR expression in the proteomic or flow cytometric studies. Thus, we used the SW620 cell line as an EGFR negative control in our experiments.

Two triple-negative breast cancer (TNBC) cell lines, BT-20 and MDA-MB-468 overexpress EGFR [59,62]. At present, TNBC patients have limited options for treatment and we expected that the EGFR-directed polyplexes might be suitable for treating TNBC patients whose tumors overexpress the EGFR. Cell surface expression of EGFR levels on these cell lines was determined. Both breast cell lines have lower levels of surface EGFR compared to the Difi and A431 cell lines. MDA-MB-468 has slightly less than half the Difi EGFR levels (48%) and the BT-20 cell line had approximately a quarter of the Difi EGFR levels (21%) (Fig 3B). Similar results have been reported for total EGFR levels using Western blot analysis (Mohan *et al.*, 2021). The BT-20 and MDA-MB-468 EGFR levels are similar to the EGFR levels on CRC cell lines C135 and Lim2405 respectively (see Fig 3A,3B). Thus, both BT-20 and MDA-MB-468 cell lines can be considered to have moderate levels of EGFR on the cell surface.

We also used three cell lines which express very low EGFR levels: BT-747 [63], MCF-7 [63] and SK-BR-3 [63]. Although we did not measure the EGFR levels on these cell lines each has been studied extensively and the flow cytometry data from Stanley, et al [63] shows they all have low EGFR levels. The other cell line used for the cytotoxicity studies MDA-MB-231 had low to moderate levels of the EGFR [63], i.e., EGFR was detectable by flow cytometry, but clearly the EGFR levels were only 5% the levels in the over-expressing cell line MDA-MB-468 [63].

### *In vitro* cancer cell killing by PPEA-polyplexes

To assess the cytotoxic activity of the PPEA-polyplex on cells in culture, we used cell lines that expressed different levels of the receptor (from undetectable to more than 10$^6$ EGFR per cell). After preparation of the PPEA-polyplex and before use, we checked the size of the polyplex using the zetasizer. The particle size of the PPEA-polyplex was 119 ± 26 nm. The PPEA triconjugate (i.e., the affibody-PEI-PEG complex without the polyIC) was used as a control for nonspecific toxicity to the cells. As illustrated in Fig 4A and 4B, the PPEA-polyIC-polyplex induced a strong killing of EGFR-over-expressing cells (1-2x10$^6$ EGFRs/cell) as compared to a cell line devoid of EGFR (U138-MG). In contrast, treatment with the PPEA, polyI-polyplex did not result in cell killing in either of the cell lines tested (Fig 4C), indicating that growth inhibitory activity induced by PPEA treatment is polyIC-dependent. PPEA-polyIC-polyplexes also produced a significant growth inhibitory effect in the SK-BR-3, BT-474 and MDA-MB-231 cell lines, which express medium levels of the receptor (1x10$^5$-3x10$^5$ EGFRs/cell) (Fig 5). These results show that PPEA-polyIC-polyplexes inhibit the proliferation of tumor cell lines with medium to high levels of EGFR expression.

Cells were treated with polyIC at the indicated concentrations using PPEA. Cell viability was measured as described in Fig 4. MDA-MB-231 cells express 2.5-3x10$^5$ EGFRs/cell, SK-BR-3 express 3x10$^5$, BT-474 express 10$^5$ EGFRs/cell and MCF7 cells express 5x10$^3$ EGFRs/cell.

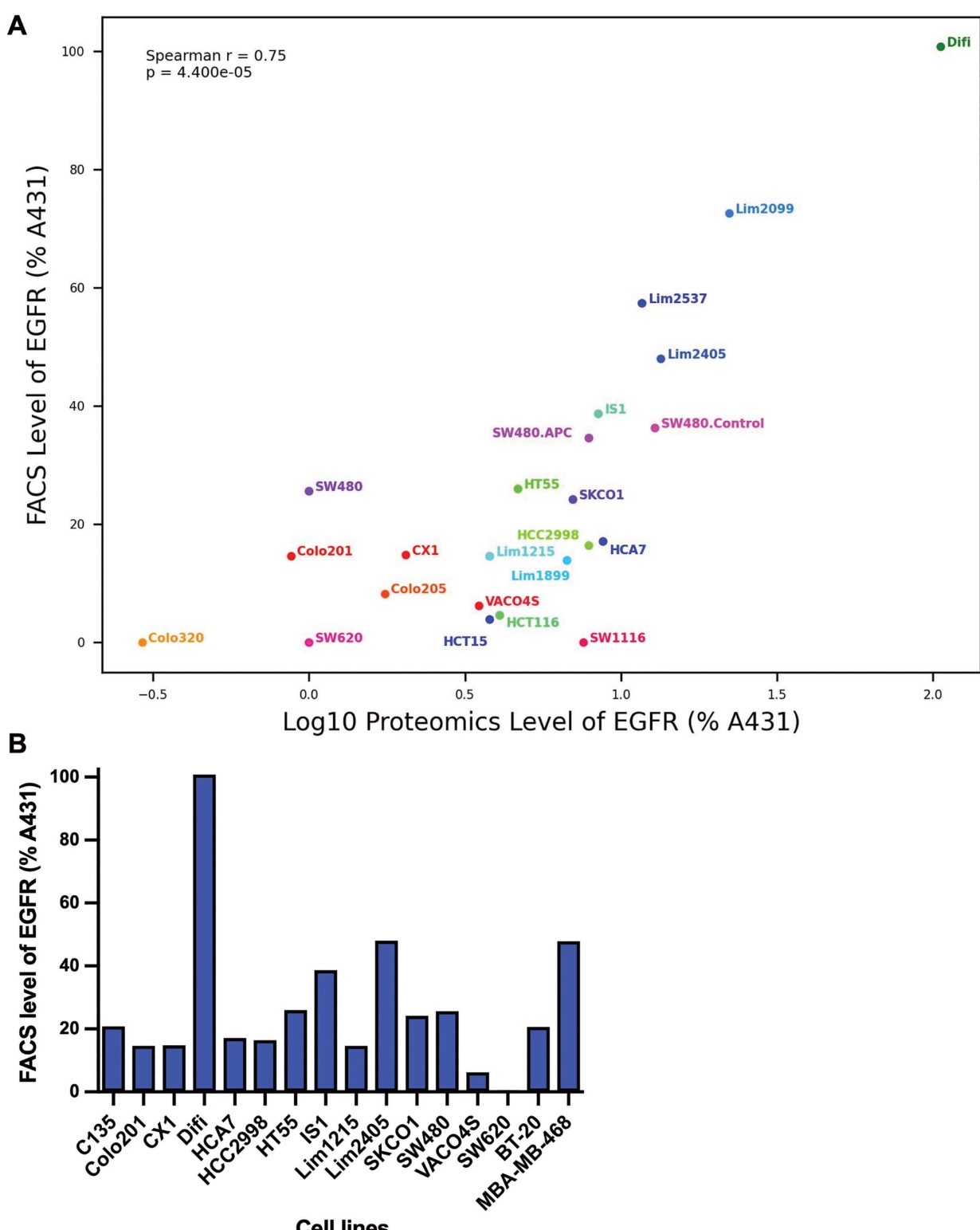

**Fig 3. Relationship between cell surface EGFR and total EGFR in cancer cell lines. (A)** Cell lines were incubated with anti-EGFR-PE and the PE fluorescence was measured by flow cytometry. The median fluorescence intensity for the respective quadrant was used to calculate relative levels of

EGFR expression on cell surface compared to the A431 cell line. Proteomic data was calculated from mass spectrometer peptide counts. **(B)** Relative levels of EGFR expression on cell surface of several CRC cell lines and breast cancer cell lines, BT-20 and MDA-MB-468, compared to the A431 cell line.

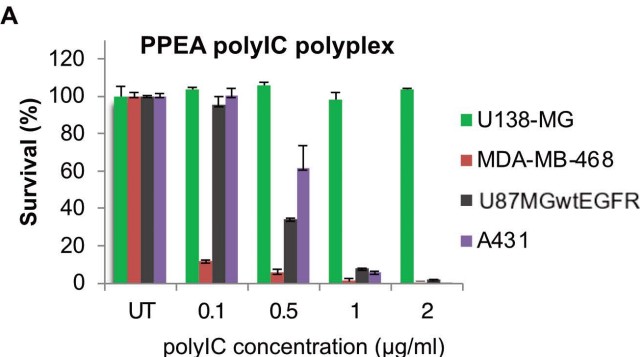

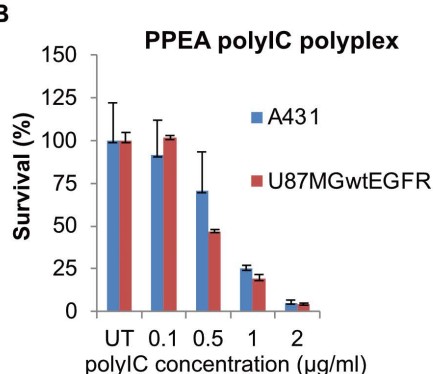

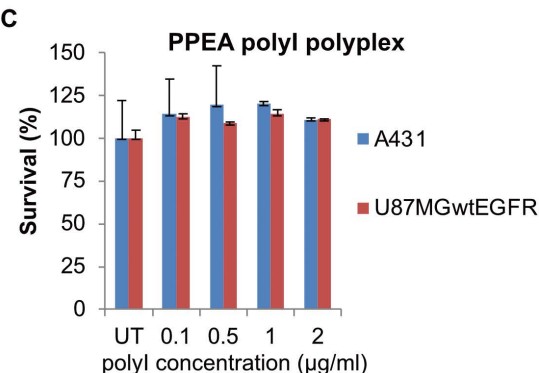

**Fig 4. Anti-tumor activity of the PPEA-polyIC-polyplex *in vitro*. (A)** PPEA-polyplexes selectively kill cell lines overexpressing EGFR. Cells were seeded in duplicates into 96-well plates at a density of 5000 cells in 0.1 ml medium per well and grown overnight. Cells were then treated with polyIC at the indicated concentrations using the PPEA complex. PEI-PEG ratio = 1:1; w/w ratio PEI: polyIC = 0.78. U138MG cells do not express EGFR; U87MGw-tEGFR cells express $1 \times 10^6$, A431 express $2-3 \times 10^6$ and MDA-MB-468 express $2 \times 10^6$ EGFRs/cell. **(B,C)** *In vitro* anti-tumor activity of PPEA-polyIC-polyplex is polyIC-specific. A431 and U87MGwtEGFR cell lines were treated with same doses of PPEA-polyIC-polyplex **(B)** or PPEA polyI polyplex **(C)**, which served as negative control. Viability was measured by the PrestoBlue Cell Viability Reagent (Invitrogen), according to the manufacturer's instructions, at 72 hrs after treatment. These experiments were repeated three times with a representative experiment shown.

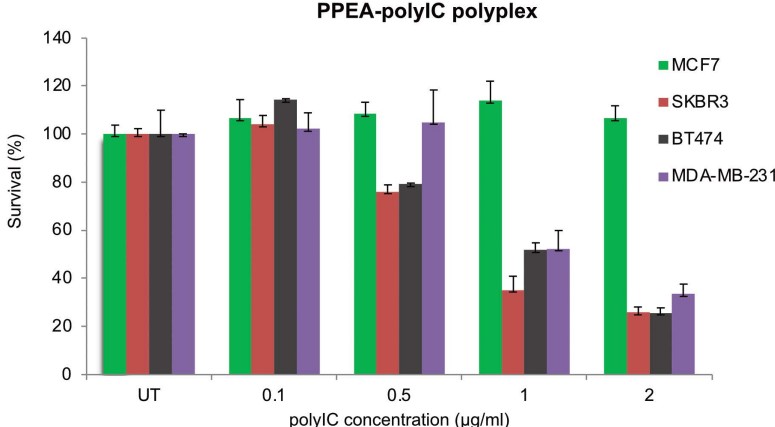

**Fig 5. Anti-tumor cell activity of PPEA-polyplex in cell lines with medium levels of EGFR.**

To investigate the relationship between the level of EGFR expression on cell surface and the potency of the PPEA-polyplex, we determined $EC_{50}$ values for two colon cancer cell lines SW480 and Difi and two breast cancer cell lines BT-20 and MDA-MB-468. Upon treatment with PPEA-polyplex (Table 2, Figure S4), the cell viability assays showed significant growth inhibition of the CRC cell lines with medium levels of EGFR overexpression such as LIM1215 (($EC_{50}$ < 370 ng/ml) Figure S4A and S4B). Despite the very high levels of EGFR on Difi cells the PPEA-polyplex was no more potent, than it was on the LIM1215 cells (Table 2, Figures S4B and S4C).

The PPEA-polyplex was a potent inhibitor of the breast cancer cell lines BT-20 and MDA-MB-468 even though they have less than half the EGFR levels of A431 cells, indeed the BT-20 cell line was 2 orders of magnitude more sensitive to PPEA-polyplex killing than the other cells (Table 2).

## PPEA-polyplex does not induce EGFR phosphorylation

The affibodies in the $Z_{EGFR1907}$ family have been reported to bind the EGFR without activating the kinase [29,30]. By treating MDA-MB-468 cell with the PPEA-polyplex and analyzing the autophosphorylation of EGFR we confirmed that the $Z_{EGFR1907}$ affibody, when tethered to PEI-PEG, does not activate the EGFR kinase (Supplementary Figure S5), i.e., there was no increased EGFR phosphorylation in the cells treated with the PPEA-polyplex.

**Table 2. Cytotoxicity of the PPEA-polyplex on selected cancer cell lines.**

| Cell Line | PPEA-Polyplex $EC_{50}$ (µg/ml) |
|---|---|
| SW620 | 1.47 ± 0.06 |
| Difi | 1.63 ± 0.07 |
| BT 20 | 0.80 ± 0.80 |
| MDA MB 468 | 1.21 ± 0.26 |

Note: The values shown on the table represent the mean values calculated with ± values representing the standard deviation (SD).

### The PPEA-polyplex induces expression of chemotactic cytokines in cancer cells expressing high and moderate levels of EGFR

Tumor cells that internalize polyIC are induced to secrete cytokines that attract immune cells [64,65]. In our previous studies, we showed that polyIC targeted by PPE, induced expression of Interferon ɣ-induced protein 10kDa (IP-10), and growth- regulated protein α (Gro-α) in EGFR-overexpressing cell lines, but not in cells devoid of EGFR [23,24]. Gro-α and IP-10 are chemokines that play an active role in the recruitment of leukocytes to the domain in which they are secreted [66,67]. To evaluate the ability of PPEA-polyplex to induce EGFR-expressing cells to secrete inflammatory cytokines, we collected cell culture supernatants 48 hrs after exposure to the PPEA-polyplex and assessed for the presence of cytokines Gro-α and IP-10. As can be seen in Table 3, cells harboring high or moderate levels of EGFR secrete chemotactic cytokines following challenge with PPEA-polyplex. In contrast, no cytokine secretion was detected in U138-MG and MCF-7 cells that are devoid of (or express low levels of) the EGFR, respectively, following treatment with the PPEA-polyplex. Interestingly, we observed significantly higher amounts of IP-10 in supernatants collected from MDA-MB-468, SK-BR-3 and BT-474 cells, compared to A431 and MDA-MB-231 (Table 3). This observation points to possible differences in signalling mechanisms mediated by PPEA-polyplex treatment in the cell lines tested. Collectively, these data demonstrate that polyIC treatment using PPEA stimulates release of pro-inflammatory cytokines by tumor cells expressing high and moderate levels of EGFR.

### PPEA-polyplex treated cancer cells activate human immune cells *in vitro*

We used peripheral blood mononuclear cells (PBMCs) to assess the ability of cytokine-enriched medium derived from PPEA-polyplex treated tumor cells to stimulate the immune cells [27]. PBMCs consist of several types of immune cells, including monocytes, macrophages, NK (natural killer) cells and T-cells. When stimulated, PBMCs produce an array of cytokines which can be conveniently quantified by ELISA [68].

We tested whether the chemotactic stimuli secreted by cells that internalized polyIC stimulated PBMC to migrate toward the secreting cells [69]. Using chemotactic chambers (see Materials and Methods) we showed that the cytokine-enriched medium from A431 cells treated with PPEA-polyplex stimulated chemotaxis of PBMCs (Fig 6). In contrast, medium of U138MG treated with PPEA-polyplex did not induce chemotaxis. Thus, PBMCs are attracted by medium from EGFR overexpressing cells, which have been treated with PPEA-polyplex. The role of polyIC would be clearer if we had used a PPEA-polyI control.

The PPEA-polyIC-polyplex selectively kills cell lines overexpressing EGFR. Cells were seeded in duplicates into 96-well plate at a density of 5000 cells in 0.1ml medium per well and grown overnight. Cells were then treated with polyIC

**Table 3. Secretion of chemokines after transfection with PPEA-polyplex.**

| PPEA-polyplex | Cytokines (pg/ml)** | | | |
| | IP-10 | | Gro-α | |
| | – | + | – | + |
|---|---|---|---|---|
| U138MG | 0 | 0 | 0 | 0 |
| MCF7 | 0 | 0 | 0 | 0 |
| A431 | 30±4 | 91±3 | 0 | 111±20 |
| MDA-MB-231 | 15±7 | 93.5±2 | 42±1 | 147±5 |
| SKBR3 | 73±13 | 634±35 | 0 | 142±21 |
| BT474 | 65±2 | 574±46 | 18±5 | 49±2 |
| MDA-MB-468 | 0 | 377±26 | 100±4 | 0 |

Cells were transfected with PPEA formulated with 2 µg/ml of polyIC (as described in Figure S4). Following 48 hours of incubation, the supernatants from each cell line were collected and subjected to ELISA determination of IP-10 and Gro-α (PeproTech). Data represent means of duplicate wells ±SD and are representative of 2–3 experiments. Standard deviations are shown in *Italic*.

**Cytokine levels measured by ELISA following 48 hours transfection using PPEA formulated with 2 µg/ml polyIC.

 

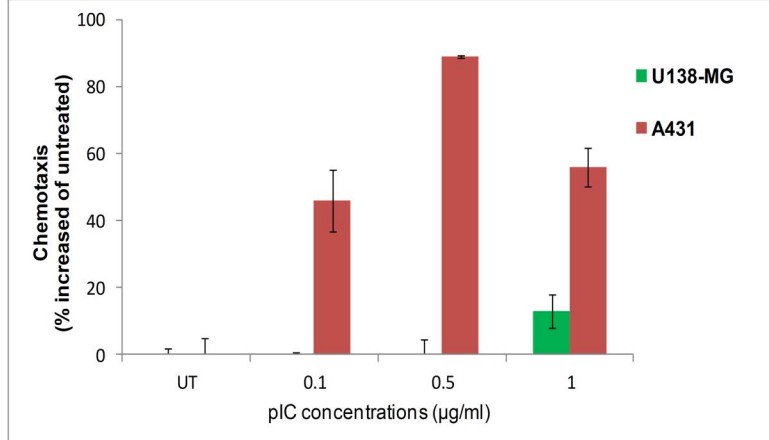

**Fig 6. Activity of conditioned media from PPEA-polyIC-polyplex treated cancer cell lines on PBMC chemotaxis *in vitro*.**

at the indicated concentrations using the PPEA complexes. PEI-PEG ratio = 1:1; w/w ratio PEI: Poly(IC) =0.78. U138MG cells do not express EGFR; A431 express $2\text{-}3\times10^6$ and MDA-MB-468 express $2\times10^6$ EGFRs/cell.

We examined the ability of the medium from PPEA-polyplex treated cancer cells to stimulate PBMCs. Upon activation, PBMC produce cytotoxic cytokines, such as INF- γ and TNF-α, known to contribute to tumor suppression and eradication [70,71]. PBMC were challenged with culture supernatant from PPEA-polyplex treated cells and their response was measured using ELISA. Table 4 illustrates the levels of INF-γ and TNF-α expression by the immune cells, detected 24 hrs following the challenge. The data clearly show that the medium originated from polyIC-targeted A431, MDA-MB-231 and SK-BR-3 cells (polyIC, 2 µg/ml) stimulate the production of cytotoxic cytokines as compared to culture supernatant from untreated cells or unchallenged PBMC cells (Table 4). We conclude that the culture supernatants derived from polyIC-targeted EGFR overexpressing cells sustain the capability to selectively stimulate the immune cells whereas cells devoid of EGFR do not.

## PBMC-mediated bystander effect

Data from clinical studies has demonstrated a correlation between the presence of tumor infiltrating lymphocytes which produce cytokines such as INF-γ and TNF-α, and better outcomes for patients with many different cancers [72,73]. Previously,

**Table 4. Evaluation of cytotoxic cytokine levels secreted by activated PBMC cells.**

| Cytokine (pg/ml)*** | | | | |
|---|---|---|---|---|
| **INF-γ** | | **TNF-α** | | |
| **+** | **−** | **+** | **−** | **PPEA polyplex** |
| 73 ± 7 | 330 ± 15 | 560 ± 26 | 98 ± 24 | A431 |
| 1 26 ± 21 | 7 ± 1 | 311 ± 58 | 71 ± 10 | MDA-MB-231 |
| 52 ± 14 | 10 ± 6 | 147 ± 9 | 57 ± 7 | SKBR3 |
| | 9 ± 2 | | 44 ± 6 | PBMCa |

MDA-MB-231, A431 and SKBR-3 cell lines were seeded in duplicates into 96-well plate at a density of 10,000 cells in 0.1 ml per well. Cells were then transfected with the PPEA-polyplex (PPEA formulated with 0.5–2 µg/ml of polyIC). Following 48 hours of incubation, 200 µl of supernatants from each cell line were collected and added to PBMCs seeded in duplicates into 96-well plate at a density of $3\times10^6$ in 0.1 ml medium per well. Following 24 hours incubation, the supernatants were collected and subjected to TNF-α and INF-γ determination using ELISA (PeproTech). Data represent means of duplicate wells ±SD and are representative of 2–3 experiments.

***Cytokine secretion measured by ELISA following exposure of PBMCs to medium from cells transfected using PPEA formulated with 2 µg/ml polyIC.

aShows expression of cytokines in PBMCs incubated with growth medium alone and served as a negative control.

we reported that expression of INF-γ and TNF-α by PBMCs strongly enhanced the bystander killing of untreated tumor cells [27]. To assess the PBMC-mediated bystander effect, MDA-MB-231 and SK-BR-3 cells were first treated with PPEA-polyplex and 24 hrs later, freshly isolated PBMCs were added to the treated cells for co-incubation. Following a further 24 hrs, medium from the treated cells co-cultured with PBMC, was added to newly seeded, untreated cells (Fig 7A). The PBMC-mediated bystander killing was examined 72 hrs later, and compared with the "direct" bystander killing, mediated by medium from PPEA-polyplex-treated MDA-MB-231 and SK-BR-3 cells (Fig 7B and 7C). The medium derived from polyIC-treated cells co-incubated with PBMCs, exerted a strong PBMC-mediated bystander effect, killing up to 80−90% of the nontreated cells (Fig 7C). Medium derived from MDA-MB-231 or SK-BR-3 cells challenged with PPEA-polyplex alone, eliminated only 20 and 35% of untreated tumor cells, respectively (Fig 7B). These results demonstrate that PPEA-polyplex treatment and immune cells display a cooperative cell-killing activity by enhancing bystander killing of EGFR-overexpressing tumors.

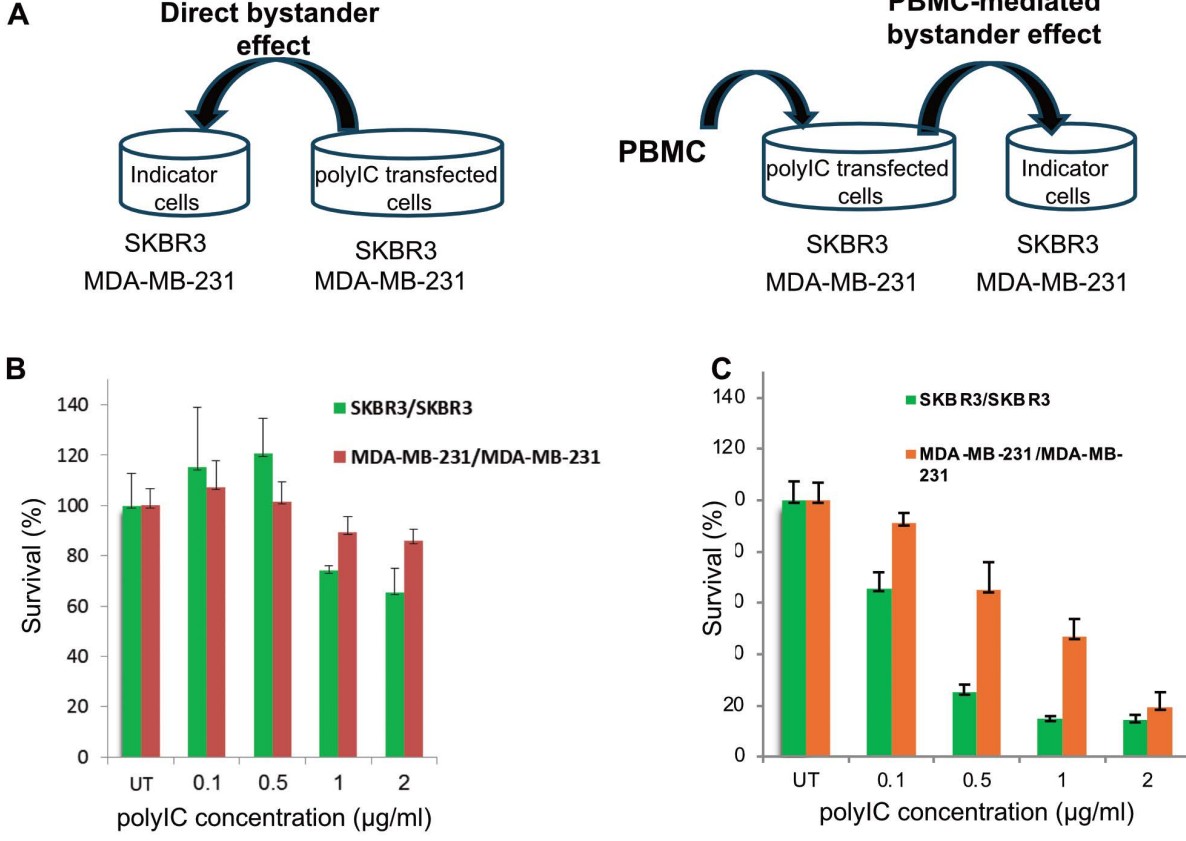

**Fig 7. "Direct" versus PBMC-mediated bystander effect.** 50,000 of MDA-MB-231 and SK-BR-3 cells were seeded into 24-well plates in duplicates and grown overnight with 1 ml medium per well. Cells were then treated with the PPEA-polyplex at the indicated concentrations. To test the "direct" bystander effect, 48 hrs following treatment 0.1 ml of medium from challenged cells was exchanged for 0.1 ml medium from non-treated cells ("indicator cells) seeded on 96 well plates (4000 cell/well) 24 hrs earlier. To analyze the PBMC-mediated bystander effect, 5x105 of PBMCs were added to treated cells 24 hrs following treatment. After 24 hrs of co-culture, 0.15 ml of medium from the "conditioned medium" was added to untreated cells seeded into 96 well plates (4000 cell/well) 24 hrs earlier. Survival of untreated cells was determined by methylene blue assay, 72 hrs after challenge with the conditioned medium. **(A)** Shows experiment design. **(B)** Shows the "direct" bystander effect of medium from polyIC-targeted MDA-MB-231 and SK-BR-3 cells on unchallenged MDA-MB-231 and SK-BR-3, respectively. **(C)** Shows PBMC-mediated bystander effect of PBMCs co-incubated with polyIC-treated MDA-MB-231 and SK-BR-3 cells on unchallenged MDA-MB-231 and SK-BR-3, respectively.

### PPEA complexes induce bystander effects

"Direct" versus PBMC-mediated bystander effects were also measured by treating MDA-MB-231(human breast cancer) and SK-BR-3 (human breast cancer) cells with the PPEA-polyplex and testing the direct or indirect bystander effects of the medium on PBMCs. Figure S6A shows the experimental design, Figure S6B shows the "direct" bystander effect of medium from polyIC- targeted MDA-MB-231 and SK-BR-3 cells on unchallenged MDA-MB-231 and SK-BR-3, respectively and Figure S6C the PBMC-mediated bystander effect of PBMCs co-incubated with polyIC- treated MDA-MB-231 and SK-BR-3 cells on unchallenged MDA-MB-231 and SK-BR-3, respectively. The supernatants from PPEA-polyplex treated cells activate human immune cells.

### Efficacy of PPEA-polyplexes on human A431 xenografts growing in nude mice

The *in vivo* antitumor activity of the PPEA-polyplex was examined using the A431 subcutaneous xenograft model. Figure 8 shows that the PPEA-polyIC-polyplex inhibited tumor growth strongly. As a control, we treated the animals with a PPEA-poly Inosine (polyI)-polyplex made in the same way as the PPEA-polyIC-polyplex. Poly Inosine is not a double stranded RNA and would not be expected to kill the cells either directly or to induce immune cytokines which amplify the killing of the tumor cells. The tumors treated with the PPEA-poly Inosine (polyI)-polyplex grew at a similar rate as tumors in the untreated group.

   Female nude mice were injected subcutaneously (s.c.) with A431 cells as described in the Methods section. After tumor establishment, mice were randomly divided into five groups and treated with daily intravenous injections (6 days/week) of the indicated polyplex at either 0.75 mg/kg or 0.1 mg/kg for 10 days, totaling 9 injections. Tumor volumes were measured on days 3, 6 and 10, and the chart presents the average tumor volume for each group. Statistical analysis: Two-tailed paired t test. Data are presented as mean ± SEM. P-values ware for comparison with the tumors growing in untreated (UT) mice; ** $p < 0.01$; *** $p < 0.001$.

## Discussion

Combining the immune system's ability to kill cancer cells together with engineered antibodies that target cancer cells has been shown to be an effective treatment of some cancers [74–78]. Signaling from the EGFR family has been reported to drive the growth of many human cancers [13,79–83]. This motivated us to target the EGFR and at the same time attempt to stimulate anti-tumor immune responses and activate killing pathways within the tumor cells. Targeted polyIC therapy offers a promising approach for improving cancer therapy [81–83]. Chemical vectors for therapeutic applications have the advantage over viral vectors since they can be custom designed to target specific cells, are less immunogenic and cannot

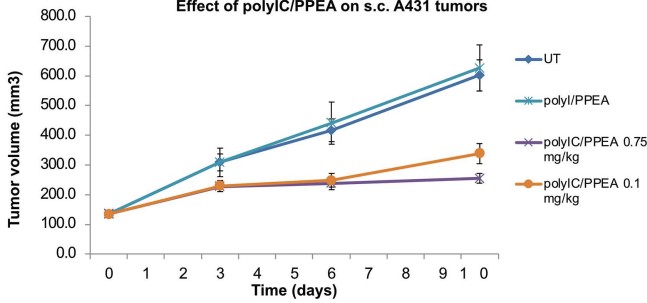

**Fig 8. *In vivo* efficacy of PPEA-polyIC and poly I-polyplexes.**

interact with the host genome. Affibodies directed against the EGFR or EGFR ligands coupled to a scaffold which can bind polyIC have the potential to deliver polyIC to cancers with elevated, active EGFR.

We produced and purified the $Z_{EGFR\ 1907'}$-affibody-polyIC-polyplex (PPEA-polyplex). The efficiency of drug delivery is influenced by the biophysical properties of the polyplex such as size, shape and surface charge [84,85]. Previous biophysical and structural studies have revealed that $sEGFR_{1-501}$ (domain I, II and III) binds to EGF ligand with greater affinity than the full-length EGFR ectodomain [86]. Cetuximab (monoclonal antibody) interacts exclusively with domain III sEGFR, hindering ligand binding and sterically preventing dimerization [53]. Analysis of the EGFR binding of the PPEA-polyplex revealed an average $K_D$ of 6.74 nM to $sEGFR_{1-621,}$ suggesting domain IV may function to stabilize binding of the affibody (Fig 2). Kinetic data models together with the significant difference in the $k_{on}$ and $k_{off}$ values suggest the presence of two binding site conformations for the $Z_{EGFR\ 1907'}$ on $sEGFR_{1-621}$, with one binding site much weaker than the other. This may be due to the presence of both tethered/non tethered forms of $sEGFR_{1-621}$ or the orientation of the $sEGFR_{1-621}$ on the sensor chip. Higher $K_D$ values obtained for $sEGFR_{1-501}$ is most likely to be due to its inability to form the untethered back-to-back conformation which occurs on dimerization and is required for tight binding of $Z_{EGFR\ 1907'}$-affibody. The high affinity binding of Cetuximab to sEGFR makes it difficult to measure the $K_D$ values using the biosensor. However, the trend in $K_D$ values suggests there was much tighter binding between Cetuximab and $sEGFR_{1-621}$ than between Cetuximab and $sEGFR_{1-501}$, indicating the need for domain IV to facilitate tight binding of Cetuximab to the $sEGFR_{1-621.}$

Saturated binding of Cetuximab followed by $Z_{EGFR\ 1907'}$ shows that $Z_{EGFR\ 1907'}$ has minimal available space to bind, whilst the opposite is true when the affibody dissociates and thus providing binding sites for Cetuximab. These findings lead us to believe that the affibody binds to the same location on the EGFR as Cetuximab – i.e., domain III.

The potency of these polyplexes was tested on a range of selected cell lines expressing different levels of EGFR: CRC cell lines, breast cancer cell lines and A431 cell line (Fig 4). In general, cells with more receptors are killed more effectively by the EGF-conjugated polyplex, however, some cells with high levels of EGFR are still resistant to the polyplex, whereas other cells with moderate levels of EGFR are sensitive to low concentrations of the polyplex. For example, the breast cancer cell lines used in this study have been reported to have only moderate levels of EGFR [59,62] yet these cell lines were shown to be the most sensitive to the polyplexes. There are several possible explanations, e.g., EGFR receptor turnover might be required, or cell survival pathways are deficient in some cells. Further investigation is required to identify biomarkers which can indicate which patient tumors are likely to respond to the polyplex. It will be important that each patient's tumor is tested for EGFR levels and that organoids from their tumor can be assayed to determine their likely responses to the polyplexes. It will be interesting to investigate whether the rate of turnover of EGFR in these cell lines plays a part in their susceptibility to the polyplex.

We tested the potency of polyIC complexed with a non-viral vector PPEA – an EGFR-homing vector which does not activate the EGFR (Fig 5). Previous studies had shown that GE11, an EGFR-homing peptide devoid of receptor activation, showed good potency in both in cells and in vivo [87]. Yet the affinity of the GE11 based vector is not sufficiently high to enable its development as a clinical candidate [32]. In contrast, the EGFR-binding affibody molecule ($Z_{EGFR1907'}$) exhibits nanomolar binding affinity to EGFR, does not activate the kinase moiety of the receptor [88,89] but its polyIC-polyplex (PPEA) kills EGFR positive tumor cells effectively.

Significantly, we found that PPEA-polyplexes induced a strong EGFR-specific killing effect in breast cancer cell lines expressing medium levels of the receptor ($1 \times 10^5$-$3 \times 10^5$ EGFRs/cell (Fig 5). It is possible that the high efficacy of polyIC delivery by PPEA, along with its inability to activate EGFR, is the origin of the efficacy of PPEA-polyplexes in cells with moderate expression of the receptor. It would be interesting to understand how the EGFR system PPEA complexes deliver the polyIC to the cytoplasm. The LPEI peptide accounts for the disruption of the endosome barrier, but why is the PPEA-conjugate internalized from the cell surface to the endosome? This may be the key to understanding why some cells with lower EGFR expression are so sensitive to the PPEA. Possible mechanisms include constant EGFR internalization in tumor cells driven by autocrine ligand (TGFα) production, internalization of EGFR hetro-oligomers driven by HER3

ligands. Once the polyIC is in the cytoplasm the killing and inflammatory responses will be activated, by the arrangement of the pathways responding to dsRNA might be significantly different in different cell lines. Measurement of the differential sensitivity of tumor cells to the EGF-based or affibody-based polyplexes will be an important for deciding which patients are likely to benefit from the use of these agents.

The ability of polyIC targeted by PPEA to induce expression of chemotactic cytokines in EGFR-overexpressing cells was comparable to that exhibited by an equivalent polyplex vector which used EGF target and activated the EGFR. More-over, the supernatant fluids from PPEA-polyplex treated cells were able to activate human immune cells and to induce production of cytotoxic cytokines.

*In vivo* antitumor activity of PPEA was demonstrated using A431 subcutaneous xenografts (Fig 8). Based on this data, the growth of the A431 tumors in mice treated with PPEA-polyplexes was delayed by more than 7 days. This effect was dependent on the polyIC, as the PPEA-poly Inosine did not reduce tumor growth. Although nude mice do not have T-cells, the innate immune system is still intact and can help to kill the tumor cells [90–92]. We considered using an allograft model to investigate the role of the immune system in the killing of tumor cells by PPEA *in vivo*, but murine cells with sustained elevated levels of EGFR were difficult to establish both *in vitro* and as allografts. Our results showed that human triple negative breast tumor cells are killed by PPEA *in vitro*; it would be interesting to establish a triple negative breast tumor cell allograft model suitable for investigating the effects of PPEA.

PolyIC is a potent activator of intracellular killing pathways as well as cytokine release [93–96]. Others have reported that PolyIC can induce the killing of human tumor cells growing in nude mice [97]; *in vivo* tumor cell killing by PolyIC can occur in the absence of T-cells. Once internalized, PolyIC is known to induce tumor cell killing, both directly via the induction of ds-RNA activated apoptosis [25] and indirectly from the release of cytokines which activate dendritic cells, NK cells, macrophages, eosinophils and neutrophils [25]. The direct killing can occur via PolyIC activating the endosomal Toll-like receptor 3 system (TLR3) [98] and/or the cytosolic melanoma differentiation-associated protein 5 system (MDA5) [99]. TLR3 is expressed in many cancer cells [100,101]. Activation of the TLR3 system not only induces the secretion of cytokines [101], but also induces a reduction in tumor cell proliferation and increased tumor cell apoptosis [102], which are likely to contribute to reduced tumor growth.

In syngeneic systems, i.e., the host and the tumor cells are the same species, not only will direct and innate tumor cell killing be induced by PolyIC, there will also be killing of the tumor cells by T-cells activated as a result of polyIC induced cytokines. The PPEA polyplex is likely to be a more potent anti-tumor agent in both in syngeneic tumor models and cancer patients [103,104].

Delivering polyIC selectively to cancer cells provides the potential for a specific anti-tumour response [96,105]. Many common cancers over-express the EGFR, by attaching polyIC to PEG and the poly-imine there is less non-specific cellular uptake; the addition of the high affinity EGFR ligand (i.e., the affibody) then directs the polyIC to the tumor cells. It was surprising to see the polyIC internalized without receptor kinase activation, so it will be intriguing to unravel the mechanism of internalization. Autocrine stimulation of EGF-HER3 oligomers is one possible mechanism, but detailed experiments will be required to discover the mechanism of the internalization of the affibody-polyplex. Once in the endosome, the LPEI component will facilitate the disruption of the membrane and the entry of the polyIC to the cytoplasm, where both the dsRNA killing mechanisms and the induction of anti-tumor cytokines are initiated.

In conclusion, PPEA polyplexes effectively stopped the growth of the A431 tumor, which has high levels of EGFR expression, from growing in nude mice. However, our studies with cancer cell lines growing *in vitro*, indicated that not all cancer cells with very high EGFR expression will be killed by the affibody-polyplex. The affibody-polyplex induced immune cytokines and promoted bystander killing of tumors *in vitro*, without activating EGFR kinase activity. We have examined the effects of the affibody on a range of cell lines, but we have not studied the effects of the affibody on primary tumor organoids or extensively in syngeneic mouse tumor models with intact immune systems. More extensive data with the organoids [106,107] and with syngeneic mouse models [108–110] will be required to strengthen case for clinical testing

of the affibody-polyplex as an anti-cancer agent. However, our initial findings underscore the therapeutic potential of the PPEA-polyplex and warrant further evaluation of this agent in preclinical and clinical anti-cancer models.

## Supporting information

**S1 Fig. Flow Chart summarising the synthesis of the Affibody-polyplex (EGFR Affibody $Z_{EGFR\ 1907}$, poly-IC-Polyplex).** The details for each step of the synthesis are described in the Materials and methods section. (TIFF)

**S2 Fig. Two binding site model for $Z_{EGFR\ 1907}$, affibody with $hEGFR_{1-621}$.** Analysis of binding data obtained for $Z_{EGFR\ 1907}$, affibody with $hEGFR_{1-621}$ provides justifications for two sites fits model. Solid and dotted line graphs represent raw and fitted data respectively. (TIFF)

**S3 Fig. Validating binding kinetics of $Z_{EGFR\ 1907}$, affibody binding to EGFR proteins. (A)** Binding kinetics for the $Z_{EGFR\ 1907}$, affibody and Cetuximab interacting with $hEGFR_{1-501}$ and $hEGFR_{1-621}$, measured using the Biacore S200 and used to determine $K_d$ values. **(B)** Analysis of binding data obtained for $Z_{EGFR\ 1907}$, affibody with $hEGFR_{1-621}$ provides justifications for two sites fits model. Solid and dotted line graphs represent raw and fitted data respectively. (TIFF)

**S4 Fig. Inhibition of cell proliferation by PPEA-$Z_{EGFR\ 1907}$, affibody-polyplex. (A)** LIM 1215 cells were treated with PPEA polyplex, polyIC, $Z_{EGFR\ 1907}$, affibody, or the cytotoxic drug WEHI-7326 for 4 days. **(B)** LIM 1215 cells were treated with PPEA polyplex, polyIC, $Z_{EGFR\ 1907}$, affibody, or the cytotoxic drug WEHI-7326 for 2 hr, then cultured for 4 days. **(C)** DiFi cells were treated with PPEA polyplex, polyIC, $Z_{EGFR\ 1907}$, affibody, or the cytotoxic drug WEHI-7326 for 2 hr, then cultured for 4 days. Cell proliferation was monitored using the CellTiter Glo assay. The cytotoxic drug WEHI-7326 was used as the positive control, i.e., WEHI-7326 induced 100% cell death. (TIFF)

**S5 Fig. Western blot analysis of autophosphorylated and total EGFR in MDA-MB-468 cells upon addition of the PPEA-polyplex.** Cells were treated as described in "Materials and methods". Untreated cells (UN) were used as negative control and cells incubated with EGF for 5 min were used as the positive control. (TIFF)

**S6 Fig. PBMC mediates bystander effects.** 30,000 A431, MDA-MB-231, MDA-MB-468, SKBR3 or BT474 cells were seeded into 48-well plates and grown overnight with 1 ml medium per well. Cells were then treated with PPEA-polyplex at the indicated concentrations. 24hrs after treatment 0.2 ml of medium from the treated cells ("conditioned medium") was added to $3x10^5$ PBMCs/well, which had been seeded immediately following isolation into 96 well plates in 0.1 ml medium/well. Following 48 hrs incubation, 0.1 ml of medium from challenged PBMCs was then exchanged for 0.1 ml medium from non-treated cells ("indicator cells) seeded on 96 well plates (4000 cell/well), 24 hrs earlier. Survival of these cells was determined by methylene blue assay, 72 hrs after challenge with the medium from the PBMCs. **(A)** Shows experiment design; **(B)** Shows the PBMC-mediated bystander effect of PBMCs challenged with medium from polyIC treated MDA-MB-231, A431 and MDA-MB-468 cells on untreated A431 cells. **(C)** Shows the PBMC-mediated bystander effect of PBMCs challenged with medium from polyIC treated SKBR3 and BT474 cells on untreated SKBR3 and BT474 cells, respectively. (TIFF)

**S1 Table. Immobilisation results using CM5 chip with one flow cell per cycle.** (TIFF)

**S2 Table. One site binding kinetics determined for Z$_{EGFR 1907}$, affibody and Cetuximab to sEGFR proteins.**
(TIFF)

**S3 Table. The Z$_{EGFR 1907}$, affibody binds to two sites on the sEGFR$_{1-621}$ protein.**
(TIFF)

## Acknowledgments

The authors wish to acknowledge Shoshana Klein for encouragement throughout this project and specifically for helping with the drafting of this manuscript. We are most grateful to Christoph Grohmann who provided expert guidance as we prepared the conjugates. Our early discussions with Donna Jovin and Thomas Jovin helped us formulate the project and they provided the initial affibody clones. Esteban Pombo-Villar at TargImmune Corporation also provided help with the development of this manuscript.

## Author contributions

**Conceptualization:** Anne Pettikiriarachchi, Antony W Burgess, Alexander Levitzki.

**Data curation:** Anne Pettikiriarachchi.

**Formal analysis:** Anne Pettikiriarachchi, Yelena Ugolev, Richard Birkinshaw, Ahmad Wardak, Alexei Shir.

**Investigation:** Anne Pettikiriarachchi, Yelena Ugolev, Alexei Shir, Nufar Edinger.

**Methodology:** Anne Pettikiriarachchi, Yelena Ugolev, Richard Birkinshaw, Ahmad Wardak, Maree C. Faux, Timothy E. Adams, Salim Joubran, Alexei Shir, Nufar Edinger.

**Project administration:** Anne Pettikiriarachchi.

**Resources:** Timothy E. Adams, Salim Joubran.

**Software:** Anne Pettikiriarachchi.

**Supervision:** Richard Birkinshaw, Ahmad Wardak, Maya Zigler, Antony W Burgess, Alexander Levitzki.

**Validation:** Anne Pettikiriarachchi, Richard Birkinshaw, Alexei Shir.

**Writing – original draft:** Yelena Ugolev, Timothy E. Adams, Alexei Shir, Maya Zigler, Antony W Burgess.

**Writing – review & editing:** Anne Pettikiriarachchi, Yelena Ugolev, Richard Birkinshaw, Ahmad Wardak, Maree C. Faux, Salim Joubran, Alexei Shir, Nufar Edinger, Maya Zigler, Antony W Burgess, Alexander Levitzki.

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
