## [Decision Letter · Decision Letter 0]

4 Nov 2025

PONE-D-25-53010EGFR-targeted affibody–polyIC polyplex kills EGFR-overexpressing cancer cells without activating the EGFRPLOS ONE

Dear Dr. Burgess,

Thank you for submitting your manuscript to PLOS ONE. After careful consideration, we feel that it has merit but does not fully meet PLOS ONE’s publication criteria as it currently stands. Therefore, we invite you to submit a revised version of the manuscript that addresses the points raised during the review process. Please make sure to address all the concerns raised by the three reviewers, especially the concerns reviewer 3 raised about the choice of the animal model and please explain how the EGFR targeted complex evaluated in this manuscript differs from or improves upon the EGF polyIC nanoparticle described in your recent publication.

We look forward to receiving your revised manuscript.

Kind regards,

Hamidreza Montazeri Aliabadi

Academic Editor

PLOS ONE

Journal Requirements:

Reviewers' comments:

Reviewer's Responses to Questions

**Comments to the Author**

1. Is the manuscript technically sound, and do the data support the conclusions?

Reviewer #1: Yes

Reviewer #2: Partly

Reviewer #3: Partly

2. Has the statistical analysis been performed appropriately and rigorously? 

Reviewer #1: Yes

Reviewer #2: No

Reviewer #3: N/A

3. Have the authors made all data underlying the findings in their manuscript fully available?

Reviewer #1: Yes

Reviewer #2: No

Reviewer #3: Yes

4. Is the manuscript presented in an intelligible fashion and written in standard English?

Reviewer #1: Yes

Reviewer #2: Yes

Reviewer #3: Yes

5. Review Comments to the Author

Reviewer #1: The manuscript presents a novel strategy employing an EGFR-targeted affibody–polyIC–polyethylenimine (PPEA) polyplex to selectively kill EGFR-overexpressing tumor cells without activating the receptor’s kinase function. The study is comprehensive, covering molecular engineering, biophysical characterization, in vitro assays, cytokine induction, PBMC activation, and in vivo xenograft validation. The authors effectively demonstrate that PPEA-polyplexes can induce potent anti-tumor and immune-stimulatory effects.

Specific comments:

1. Introduction

The narrative could be streamlined to better highlight the scientific gap this work fills compared to prior EGF–polyIC or GE11–polyIC systems. Explicitly contrast why the affibody approach improves selectivity or safety.

2. Materials and Methods

Provide a concise schematic or flow diagram summarizing the multi-step synthesis of LPEI-PEG-EGFR affibody conjugates; this would clarify the logic of the polymer chemistry.

Clarify the replication and statistical methods for all assays.

3. Discussion

Provide a mechanistic model summarizing how PPEA-polyplexes induce selective cytotoxicity and immune activation without receptor phosphorylation.

Discuss potential off-target effects or safety concerns, particularly regarding systemic cytokine release or nonspecific polyIC toxicity.

4. References

Try to cite some newly published articles, such as https://doi.org/10.1016/j.agrcom.2024.100044 , https://doi.org/10.1016/j.enceco.2025.08.016 and https://doi.org/10.1186/s12864-025-11418-y

Verify the consistency of formatting, particularly for DOI inclusion and journal titles.

5. Language and Grammar

Minor grammatical corrections are needed.

Ensure consistent terminology.

Reviewer #2: The manuscript “EGFR-targeted affibody–polyIC polyplex kills EGFR-overexpressing cancer cells without activating the EGFR” attempts to show the cytotoxic effects of this polyplex on cancer cells; however, the manuscript requires refinement and additional experiments.

Major concerns

1. The authors skip around with what cell lines they use for different assays. There is no consistency. This significantly lowers the impact of the paper. Also, sometimes only 1 cell line is used. It leaves one to assume that the experiment only works in that one cell line. Please include experiments with at least 3 consistent cell lines.

2. EGFR levels should be shown for all cancer cell lines used in other assays. Especially since it is important to the interpretation of the results.

3. You need to statistically analyze the appropriate experiments (e.g., proliferation assays, tumor growth, etc.) and a statistics section to your Materials & Methods.

4. The authors refer to low, moderate, and high EGFR expressing cells. How is this defined? What are the cut-off values?

5. The authors state “…culture supernatants derived from polyIC-targeted EGFR overexpressing cells sustain the capability to selectively stimulate the immune cells whereas cells devoid of EGFR do not.” What cells in this experiment were devoid of EGFR?

6. For the section “PBMC-mediated bystander effect” the authors refer to transfected and untransfected cells. It should be treated and untreated. Transfection is the process of introducing foreign nucleic acids (DNA or RNA), proteins, or other molecules into cells, usually to study gene function or protein expression.

7. Is the affibody murine? If it is this should be discussed for possible toxicity.

8. PolyIC given systemically is toxic. Have others shown it is less toxic when complexed with targeting moieties? This should be included in the discussion.

9. What allograft material was used? This is mentioned in the Materials and Methods but not in the results.

10. In the animal experiment results you use “PPEA-poly Inosine” with no introduction.

11. The in the results the authors state, “Human BT-20 cells would be expected to be killed in xenografts, but the immune deficient mice do not provide the immune responses which are likely to occur in patients.” What evidence do you have to support this? There are murine breast cancer cell lines that could be used.

12. The authors show in vivo tumor growth with A431 xenografts but claim to have efficacy against triple negative breast cancer. Please discuss or revise this. Why make that claim if you used a uterine cell line?

13. Please discuss the limitations of your study.

Minor concerns

14. When describing Supplementary figures sometimes an S is used in front of the number, sometimes it is not.

15. On Figure 4 and S2C the x-axis cannot be seen.

16. “MilliQ” should be replaced with deionized water.

17. Define sEGFR.

18. In the Materials and Methods, not all cell lines list their origin (e.g. breast cancer). Also, not all cell lines grown are indicated which media they were grown in.

19. For the cell viability assay, please include a supplementary table with the cell densities used.

20. The authors list bortezomib and WEHI-7326 in the Materials and Methods but they do not appear in the results. Are all the cell lines used sensitize to these drugs?

21. In the “Treatment of Subcutaneous Xenografts and Allografts” section of the Materials and Methods section, the authors state, “All 40 euthanized reached human Endpoints” and “All animals were sacrificed before reaching human endpoints.” This is confusing, please clarify. Also, add the e to the end of humane.

22. Please standardize your bar graphs in terms of size and axis.

23. Some tables need to be adjusted to avoid wrapping or displaced text.

Reviewer #3: In this study, the authors synthesize and characterize an EGFR affibody-polyIC complex (PPEA polyplex) designed to deliver polyIC to EGFR expressing cells. The rationale includes selective targeting of tumor cells with high EGFR expression and limiting the toxicity of polyIC, which is too toxic for systemic administration. The PPEA polyplex evaluated in this manuscript is similar to an EGF polyIC complex evaluated in a previous study. The authors demonstrated that EGFR expression on target cells is necessary for PPEA polyplex cytotoxicity, however the level of EGFR expression did not correlate with EC50 values. In vitro cytotoxicity assays demonstrated that the growth inhibitory activity of the complex is polyIC-dependent. The authors also showed that treatment with the PPEA polyplex resulted in secretion of cytokines and increased chemotaxis of PBMCs, suggesting an immune mediated anti-tumor response. Additionally, they demonstrated that the secretion of soluble factors and the presence of PBMCs were necessary to kill bystander cells.

The authors have previously described that polyIC binds to pattern recognition receptors to lead to the secretion of pro-inflammatory cytokines to activate both innate and adaptive immunity. The in vitro data in this current manuscript also supports the role of an anti-tumor immune response. However, they chose to use nude mice, which lack an immune system, to perform their in vivo experiments. A better model system would be a syngeneic mouse model, which allows for contribution of the native immune response. Furthermore, the in vivo data demonstrates a non-immune mediated anti-tumor mechanism. Since PPEA polyplex cytotoxicity is not dependent on EGFR expression level and the polyIC is required for cytotoxicity, the mechanism by which the complex kills tumor cells is not clear. Additionally, multiple experiments done in a 96 well plate format were only performed in duplicate. Finally, there are several errors throughout the text and many of the tables and figures are not publication quality. While the work seems promising, it is not yet ready for publication, and it is unclear how the EGFR targeted complex evaluated in this manuscript differs from or improves upon the EGF polyIC nanoparticle described in their recent publication.

Specific Comments:

• Methods: Please replace “FACS analysis” with flow cytometric analysis since cell sorting was not done. FACS is an acronym for fluorescence-activated cell sorting.

• Tables need to be reformatted. Parts of numbers and words are on different lines, text is not within the table, and values are not aligned.

• What is the difference between the data presented in Figure 2C and Supplemental Figure 2B?

• Figure 3: Only flow cytometry was performed, not cell sorting, so FACS is being used incorrectly.

• Recommend including the EGFR expression as measured by flow cytometry alone (as was done in Figure 2B for three of the cell lines) for all the cell lines included in Figure 3A. The correlation between surface and total EGFR expression from prior proteomic analysis as represented in Figure 3A is less important and can be moved to supplemental data.

• Figure 3B incorrectly states that BT-20 and MDA-MB-468 are colorectal cancer cell lines.

• Figure 4A: Recommend changing U87-MG-WT to U87-MG-WTEGFR to make it clearer that this cell line does express EGFR and to be consistent with Figure 4B.

• The overall quality of Figure 4 should be improved, including maintaining consistency in the sizes of the bar graphs across the figure.

• “In contrast, treatment with the PPEA, polyI-polyplex did not result in cell killing in either of the cell lines tested (Figure 4C), indicating that growth inhibitory activity induced by PPEA transfection is polyIC-dependent. PPEApolyIC-polyplexes also produced a significant growth inhibitory effect in the SK-BR-3, BT-474 and MDA-MB-231 cell lines, which express medium levels of the receptor (1x105-3x105 EGFRs/cell) (Figure 5).” and “Upon treatment with PPEA-polyplex (Table 2, Figure S3), the CellTiter Glo assays showed significant growth inhibition of the CRC cell lines with medium levels of EGFR overexpression such as LIM1215 (EC50 < 300 ng/ml).” The data shown here is not sufficient to demonstrate that the PPEA poly IC polyplex inhibits cell growth since cell viability was only measured at one time point.

• Table 3. Why was the U138-MG-WTEGFR cell line not used in this experiment? Comparison between cytokine levels in media from U138-MG and U138-MG-WTEGFR cells treated with PPEA-polyplex can more directly show that EGFR expression is needed to induce secretion of IP-10 and Gro-α. Recommend adding this data to Table 3.

• Figure 7. Similar to the question above, why were A431 cells and not U138-MG-WTEGFR cells used for this experiment? From the figure legend, it seems like U138-MG-WTEGFR was included, but this data is not in the figure.

• Figure 7. The title “Comparison of anti-tumor activity by PPEA-polyIC-polyplex on PBMC chemotaxis in vitro” is misleading since chemotaxis, not anti-tumor activity is being compared. Recommend rephrasing.

• Table 3 and Table 4: Cytokine assay negative controls were cells not treated with PPEA polyplex. To show this effect is polyIC-dependent, a better negative control would be the PPEA-polyI-complex used in Figure 4C.

• Figure 7 and Supplemental Figure S4: These data demonstrate that PBMCs are required for efficient cell killing and that direct cell-cell contact is not required, suggesting secreted factors are mediating the effect. Recommend moving the SKBR3/SKBR3 and BT474/BT474 data from Figure S4 to Figure 7 to better illustrate the mechanism is via secreted factors.

• Since polyIC functions by inducing an anti-tumor immune response, in vivo studies should be performed with an immunocompetent mouse model. For example, 4T1 is a syngeneic model similar to human triple negative breast cancer that could be used.

• Cetuximab is referred to as Erbitux once in the results section and once in the discussion. Including the brand name of the antibody is not necessary. Recommend removing Erbitux and only referring to the antibody as cetuximab to be consistent throughout the manuscript.

• Discussion: “Significantly, we found that PPEA-polyplexes induced a strong EGFR-specific killing effect in breast cell lines expressing medium levels of the receptor…” Should this read “breast cancer cell lines”?

• Please include line and page numbers in the future to better facilitate review of the manuscript.

• The manuscript needs to be edited prior to future submissions. There are several typos, a change in font size in the middle of the methods section, and several sentences for which phrases can be improved to be more consistent with a scientific manuscript.

6. PLOS authors have the option to publish the peer review history of their article (what does this mean?). If published, this will include your full peer review and any attached files.

Reviewer #1: No

Reviewer #2: No

Reviewer #3: No

---

## [Author Response · Author response to Decision Letter 1]

15 Jan 2026

Author Responses to the Editor and Reviewer Comments

Responses to Editor’s comments

“Please make sure to address all the concerns raised by the three reviewers, especially the concerns reviewer 3 raised about the choice of the animal model and please explain how the EGFR targeted complex evaluated in this manuscript differs from or improves upon the EGF polyIC nanoparticle described in your recent publication.”

We have responded in detail to all the reviewers’ comments, explaining our use of the animal model and how the Affibody might have advantages over the EGF based polyplex.

“Please ensure that your manuscript meets PLOS ONE's style requirements, including those for file naming.”

We have followed the guidelines.

We will use the recommended form for our Data availability statement:

“All data are in the manuscript and/or supporting information files."

“You must submit original gel images”

We have included a file with the original gel images, one of the original images ( for the kinase

Blot, now S5 fig.) has been lost in the computer filing system. This figure was supporting information for similar data published previously by others. We have referenced their publications which show similar results to our own blot.

“ Please include your full ethics statement in the ‘Methods’ section of your manuscript file”

The ethics statement is now included in the Materials and Methods section.

“Supporting Information”

Captions for our Supporting Information are at the end of our manuscript.

There was a recommendation to cite specific previously published works, but that work did not seem relevant to our data, so we did not cite it. However, as a result of the reviewers’ comments, here was a need for several new citations and these have been included.

Responses to Reviewers’ Comments

Reviewer 1

Point 1 We agree with the reviewer and we have revised the discussion to emphasize the differences between the affibody-based and EGF-based reagents.

Point 2 a concise schematic or flow diagram summarizing the multi-step synthesis of LPEI-PEG-EGFR affibody conjugates – Now included as Supplementary Figure S1

Point 3 Provide a mechanistic model summarizing how PPEA-polyplexes induce selective cytotoxicity The following paragraph has been added to the discussion:

“PolyIC is a potent activator of intracellular killing pathways as well as cytokine release(88-91). Delivering polyIC selectively to cancer cells provides the potential for a specific anti-tumour response (91,92). Many common cancers over-express the EGFR, by attaching polyIC to PEG and the poly-imine there is less non-specific cellular uptake; the addition of the high affinity EGFR ligand (i.e. the affibody) then directs the polyIC to the tumor cells. It was surprising to see the polyIC internalized without receptor kinase activation, so it will be intriguing to unravel the mechanism of internalization. Autocrine stimulation of EGF-HER3 oligomers is one possible mechanism, but detailed experiments will be required to discover the mechism of the internalization of the affibody-polyplex. Once in the endosome, the LPEI component will facilitate the disruption of the membrane and the entry of the polyIC to the cytoplasm, where both the dsRNA killing mechanisms and the induction of anti-tumor cytokines are initiated.”

Point 4 Use more recent citations: We have added several recent citations which report the anti-tumor activity of polyIC. These references should give readers a pathway to exploring more about the potential of selectively delivered polyIC-complexes as anti-tumor agents.

Reviewer 2

Point 1 The reviewer makes the point that we have used a variety of cell lines to study the different aspects of this project. In many cases the cell lines were chosen to allow us to study a the effects of the polyplexes on cells with different levels of EGFR, in others it was to measure immune responses and cell killing. We have described the use of each cell line clearly and at no stage did we select a result because it was the only cell which responded. The assays were performed in triplicate and with the large number of different cell lines we investigated for the assays, we believe the diversity and range add to the robustness of the results. Adding even more cell lines is unlikely to alter the conclusions to be drawn from our results.

Point 2 The reviewer makes a valid point. We did not measure and report the EGFR levels for four of the cell lines MCF-7, MDA-MB-231, SK-BR-3 and BT-747. We have added a paragraph to the results which now reports the EGFR levels for these cell lines. “We also used three cell lines which express very low EGFR levels: BT-747(61), MCF-7(61) and SK-BR-3(61). Although we did not measure the EGFR levels on these cell lines each has been studied extensively and the flow cytometry data from Stanley, et al(61) shows they all have low EGFR levels. The other cell line used for the cytotoxicity studies MDA-MB-231 had low to moderate levels of the EGFR(61), i.e. EGFR was detectable by flow cytometry, but clearly the EGFR levels were only 5% the levels in the over-expressing cell line MDA-MB-468 (61).” Reference 61 is a paper by Stanley et al, 2017 which documents the receptor levels for many cell lines

Point 3 Stats for appropriate experiments Where possible we have added stats for appropriate experiments.

Point 4 We have changed the title of the section” Abbreviations” to “Abbreviations and Terminology”. In this section we describe or definitions of low, moderate and high levels of EGFR on cell lines.

Point 5 Several cell lines have undetectable levels of EGFR including SW620, U138MG and MCF-7 cells

Point 6 The reviewer is correct: we have replaced “transfected” with “treated” throughout the manuscript.

Point 7 The affibody is a protein purified from bacterial extracts. We know it binds with high affinity to the human EGFR, but we did not investigate its binding to the murine EGFR.

Point 8 In the absence of polyIC, we did not observe toxicity from the affibody in mice.

Point 9 The author is correct the heading for the animal experiment included Allografts. We do not present the results of any allograft experiments in the paper, so “and Allografts” has been removed for the sub-heading in the Materials and Methods section

Pont 10 In response to the reviewer’s suggestion, we have added “As a control, we treated the animals with a PPEA-poly Inosine (polyI)-polyplex, made in the same way as the PPEA-polyIC-polyplex. Poly Inosine is not a double-stranded RNA and would not be expected to either kill the cells directly or to induce the immune cytokines which amplify the killing of the tumor cells”

Points 11 and 12 The reviewer is correct we have extrapolated too far from our results. Our abstract and the discussion of the xenograft result have been revised accordingly.

Point 13 The limitations of the in vitro data and the mouse model are pointed out in the new discussions.

Point 14 All of the supplementary figures are now referred to as S1, S2, … etc

Point 15 On Figures 4 and S2C the x-axes cannot be seen. These figures have been revised – the x-axes are now visible

Point 16 “ MilliQ” has been replaced with “de-ionized water”.

Point 17 The abbreviation sEGFR is now explained in the” Abbreviations and Terminology “ section.

Point 18 The reviewer notes: In the Materials and Methods, not all cell lines list their origin (e.g. breast cancer). We have revised the manuscript so when a cell line is first mentioned the tissue of origin is clear. Also the materials and methods includes the sentence “. Also we note that the materials and methods specifies “All cell lines were grown in their recommended media (DMEM from Gibco, RPMI 1640 from Sigma and EMEM from ATCC) supplemented with 10%FCS, -/+ Adds (1.08% thioglycerol, 50 mg/ml hydrocortisone, 100 U/ml insulin), -/+ G418.”

Point 19. For the cell viability assay, please include a supplementary table with the cell densities used. This density data is now included in the Materials and Methods

Point 20 The reviewer correctly notes that Bortezomib is listed as one of two general cell death reagents in the Materials and Methods. Actually in this manuscript, only WEHI-7326 was used and the results are documented in Supplementary Figure S3. All cell lines tested in this study were killed by WEHI-7326. Bortezomib has been removed from the ” Materials and Methods” section.

Point 21 The Ethics statement re euthanasia has been re-written

Reviewer #3

General Comments

We have added comments to the introduction and discussion to emphasize the difference between the EGF-based polyplex and the affibody-based polyplex. In particular, that the tumor killing by the affibody does not require that it activates the EGFR kinase. The observation that the killing of cell lines is dependent on the presence of the EGFR, but that the sensitivity is not related to the cell surface EGFR levels is new. This observation is likely to be important for predicting which patients are most likely to respond to these EGFR directed polyIC polyplexes. We do not have the data to determine the mechanism underlying the extreme sensitivity of some of the cell lines (e.g. the BT-20 Breast cancer cells), but in the discussion we present several possible models, including autocrine activation, EGFR hetero-oligomerization with the erbB3 system as well as alterations in the polyIC killing pathways. Significant resource, experimentation and time will be required to determine which mechanisms are responsible and will form the basis of future studies, but they are beyond the scope of this report.

Where necessary we have revised the Tables and Figures to bring them to publication quality and in accord with the PlosOne requirement.

As requested we have revised the manuscript to improve nomenclature consistency, remove typographical errors and mistakes in grammar.

Specific comments

Point 1 FACS has been replaced by Flow cytometric throughout the manuscript

Point 2 We have tried to ensure the Table formatting is robust and that text does not wrap around

Point 3 The reviewer is correct Figure 2C and Supplemental Figure S2B were the same. We have removed Supplemental figure S2B and made the appropriate changes in the figure legend and the text in the main manuscript.

Point 4 The term “FACS” has been removed from the figure and the term Flow cytometry is now used in the legend for Figure 3

Point 5 Reviewer: “Recommend including the EGFR expression”. The cell surface EGFR is almost certainly the critical data, but the total levels of EGFR may be an important indicator of the mechanism underlying the different sensitivities of the cell lines to PPEA. We prefer to leave that data in the manuscript figures, rather that relegate it to the supplementary data.

Point 6 Reviewer: “ Figure 3B incorrectly states…”, the reviewer is correct the figure has been corrected

Point 7 As recommended for Figure 4A, U87MGWT has been changed to U87MGWTEGFR

Point 8 The reviewer makes the point “The data shown here is not sufficient to demonstrate that the PPEA poly IC polyplex inhibits cell growth since cell viability was only measured at one time point”. The reviewer is correct, indeed the assay as presented only presents a value proportional to the cell number – there may have been both proliferation and cell death, inhibition of proliferation or induction of cell death. Previous experience with the polyIC system, other assays at multiple times allows us to believe that the number of cells in the population is reduced in the presence of the PPEA, thus it is reasonable to describe the effects of PPEA as growth inhibition. The underlying reason for the growth inhibition is almost certainly a combination of induced apoptosis and reduced proliferation, but we chose not to dissect the causes in this study.

Point 9 The experiment suggested by the reviewer would add definitively to the data, but the cells were generated in Melbourne and the cytokine assays were performed in Jerusalem. Exchanging the materials and expertise for this experiment would have been difficult and time consuming. We believe overall the data in our manuscript is sufficient support for the concept that the PPEA requires the EGFR for its cytokine inducing activities.

Point 10 “Why were A431 cells and not U138-MG-WTEGFR cells used for this experiment?” Similarly to point 9 – we did not exchange the U87MGWTEGFR cells between the teams. Only the U87MG cell were used in this experiment. The legend to Figure 7 (now Figure 6) has been corrected, i.e. the U87MGWTEGFR cells have been removed.

Point 11 The reviewer is correct, the title has been redrafted appropriately

Point 12 Table 3 and Table 4 . The reviewer is correct, we can only say that the PPEA-polyplex was required for inhibiting the cells, not the polyIC or PPEA independently. We have altered the discussion to make this clear. Again the reviewer is correct, if we had used the PPEA-polyplex polyI in this experiment, we could be more definitive about the role of polyIC.

Point 13 “Figure 7 and Supplemental Figure S5:” recommends moving data from S5 to Figure 7.

The supplementary figure S5 (now S6) has three parts and would overwhelm the chemotaxis data (now Figure 6). We think the results are clearer when Figure 6 only shows the chemotaxis data, so we have not combined these figures.

Point 14 The reviewer suggests we should use an immunocompetent mouse model, A431 cells growing in a nude mouse model is not ideal, especially when the these mice lack the appropriate T-cell responses. Of course, the reviewer is correct, in an ideal world we should have used a mouse tumour cells, which express medium or high levels of the EGFR, growing in a mouse syngeneic host. The editor also requested that we should provide a response about the suitability of this mouse model However, mouse tumour cells do not often express high levels of the EGFR, e.g. the 4T1 cells referred to by the reviewer only express low levels of the mouse EGFR (Nisticò, N.; Aloisio, A.; Lupia, A.; Zimbo, A.M.; Mimmi, S.; Maisano, D.; Russo, R.; Marino, F.; Scalise, M.; Chiarella, E.; et al. Development of Cyclic Peptides Targeting the Epidermal Growth Factor Receptor in Mesenchymal Triple-Negative Breast Cancer Subtype. Cells 2023, 12, 1078. https://doi.org/10.3390/cells12071078). Using the - mouse colon tumor MC38 (Tan, M.H., Holyoke, E.D. and Goldrosen, M.H., 1976. Murine colon adenocarcinomas: methods for selective culture in vitro. Journal of the National Cancer Institute, 56(4), pp.871-873, which has low levels of mouse EGFR, we made several MC38-based cell lines which expressed medium or high levels of human EGFR-GFP. These cell lines did grow in syngeneic mice (C57Bl/6), but these cells cause ulceration which interfered with the tumor growth experiments. A further complication was associated with the loss of expression of the human EGFR-GFP from the MC38 cells when they were passaged. We agree a syngeneic mouse tumor model where the tumor cells express medium or high levels of the mouse EGFR, should give more information on the potential of affibody-polyplex as a cancer therapeutic, but a new mouse model needs to be developed. It is possible that 4T1 cells would be suitable for creating cell lines which express human EGFR-GFP and which would then grow in syngeneic mice, but that would require significantly new experimentation. It is still not clear that stable expression of the human EGFR could be achieved in that cell line. If the mouse EGFR was overexpressed in mouse cells, it would be necessary check that the affinity of the affibody for the mouse EGFR would be high enough for targeting. Our difficulties with the syngeneic mouse model involved considerable experimentation, but in the end the results did not improve our knowledge of the efficacy of the affibody-polyplex.

Point 15 “Cetuximab is referred to as Erbitux” – recommend only use cetuximab not Erbitux. We have made this change

Point 16 The reviewer is correct “killing effect in breast cell lines” should be changed to “killing effect in breast cancer cell lines” –we have made this change.

Point 17 Please include line and page numbers. We have done this for the revised version of the manuscript

Point 18 The manuscript needs to

---

## [Decision Letter · Decision Letter 1]

2 Feb 2026

PONE-D-25-53010R1EGFR-targeted affibody–polyIC polyplex kills EGFR-overexpressing cancer cells without activating the EGFRPLOS One

Dear Dr. Burgess,

Thank you for submitting your manuscript to PLOS ONE. After careful consideration, we feel that it has merit but does not fully meet PLOS ONE’s publication criteria as it currently stands. Therefore, we invite you to submit a revised version of the manuscript that addresses the points raised during the review process.

Specifically, please be sure to address all the concerns raised by reviewer 3.

We look forward to receiving your revised manuscript.

Kind regards,

Hamidreza Montazeri Aliabadi

Academic Editor

PLOS One

Journal Requirements:

Reviewers' comments:

Reviewer's Responses to Questions

**Comments to the Author**

1. If the authors have adequately addressed your comments raised in a previous round of review and you feel that this manuscript is now acceptable for publication, you may indicate that here to bypass the “Comments to the Author” section, enter your conflict of interest statement in the “Confidential to Editor” section, and submit your "Accept" recommendation.

Reviewer #1: All comments have been addressed

Reviewer #2: (No Response)

Reviewer #3: (No Response)

2. Is the manuscript technically sound, and do the data support the conclusions?

Reviewer #1: Yes

Reviewer #2: Yes

Reviewer #3: Partly

3. Has the statistical analysis been performed appropriately and rigorously? 

Reviewer #1: Yes

Reviewer #2: No

Reviewer #3: Yes

4. Have the authors made all data underlying the findings in their manuscript fully available?

Reviewer #1: Yes

Reviewer #2: Yes

Reviewer #3: Yes

5. Is the manuscript presented in an intelligible fashion and written in standard English?

Reviewer #1: Yes

Reviewer #2: Yes

Reviewer #3: Yes

6. Review Comments to the Author

Reviewer #1: I would like to thank authors for their effort. I believe this manuscript has met the publishing standards.

Reviewer #2: 1. The figures I received with this submission are pixelated and in some cases impossible to read.

2. The authors have not included statistical analyses of their cell viability assays or a description of the statistics used in the animal studies.

Reviewer #3: In this revised manuscript, the authors have provided additional text in the introduction to explain the rationale for using an affibody-based PPEA polyplex in addition to their prior work that utilized an EGF-based complex. Figure quality and readability have been improved, and several typos and grammatical errors have been corrected. The also stated that they attempted to develop a murine syngeneic model that overexpressed human EGFR, however there were technical difficulties with this system.

While the manuscript is improved from the original version, several issues have not been addressed. The biggest concern that remains is that the authors present in vitro data that the PPEA polyplex induces anti-tumor activity via an immune-mediated mechanism. However, they used an in vivo model that lacks an immune system, which demonstrated that PPEA polyplex induces anti-tumor activity via a non-immune mediated mechanism. The authors added additional text to the manuscript regarding possible mechanisms of EGFR internalization, however, do not comment on the mechanism of cell killing observed in their in vivo model. The challenges of a syngeneic model including low expression of EGFR on murine syngeneic cell lines and lack of validation of the affibody to recognize murine EGFR are recognized. However, the authors would need to discuss why they observed anti-tumor activity in nude mice or use an alternative immunocompetent model, such as humanized mice. As the manuscript stands, their in vitro and in vivo data appear to be contradictory.

Additionally, while the authors present data that show differential response to PPEA polyplex based on EGFR expression, they only present data for EGFR surface expression for 3 of the cell lines used in the study, despite requests from two reviewers to provide data for additional cell lines. Instead, they provided references to other publications. Particularly since so many cell lines were used throughout the manuscript, it would be helpful to either have an immunoblot showing total EGFR expression or flow cytometric data showing surface EGFR expression comparing the cell lines directly. Additionally, there was concern from reviewer 1 that different cell lines were used for different assays. The reviewers respond to this point that they have performed experiments in triplicate to the results are robust, however some of their experiments are performed in duplicate based on figure legends and methods. The authors also stated they are unable to repeat experiments in certain cell lines due to different labs being involved in the project, which does not sufficiently address some of the reviewers’ comments. Lastly, since their point-by-point response did not include the reviewer comments, I was unable to fully ascertain if all comments from reviewers 1 and 2 were addressed.

Specific comments:

• Page 21 lines 14-17: “By treating MDA-MB-468 cell with the PPEA-polyplex and analyzing the autophosphorylation of EGFR (Supplementary Figure S5) we confirmed that the ZEGFR1907’ affibody, when tethered to PEI-PEG, does not activate the EGFR kinase (Supplementary Figure S5).” Supplementary Figure S5 does not need to be referenced twice in the same sentence. Also, the structure of the sentence can be improved to better illustrate the conclusion of the data shown in the figure.

• Table 3. The authors demonstrate that cell lines with high EGFR expression secrete the chemokines IP10 and Gro-α. In the results section of the text (page 22 line 12), the authors incorrectly state that these are pro-inflammatory cytokines.

• Page 23 lines 22 and 29, Table 4. The authors also incorrectly use the term cytotoxic cytokines when referring to pro-inflammatory cytokines. IFNγ and TNFα are pro-inflammatory cytokines. The term cytotoxic cytokines was also incorrectly used in the discussion (page 20 line 2).

• The formatting of the tables at the end of the manuscript still makes them difficult to read. It is unclear if this is consequence of the merged data file or if the authors have not corrected the formatting. The tables that are inserted within the text are better formatted.

• Thank you for changing FACS to flow cytometric analysis. However, flow is capitalized throughout the manuscript. Capitalization is not necessary and “Flow” should be changed to “flow.”

• Several typos still remain in the manuscript, including several instances of the use of the word transfection instead of treatment.

7. PLOS authors have the option to publish the peer review history of their article (what does this mean?). If published, this will include your full peer review and any attached files.

Reviewer #1: No

Reviewer #2: No

Reviewer #3: No

---

## [Author Response · Author response to Decision Letter 2]

19 Mar 2026

Author responses to the Reviewers’ comments on the first revision of our manuscript

Reviewer #1

Was satisfied with our revision of the manuscript.

Reviewer #2

1. Indicated that some of our figures in the .pdf were pixelated and impossible to read. We have fixed this issue and believe all of the figures are now clear and easy to read.

2. We have attended to the problems with the statistics – all the details of the cell viability assay statistics are now included.

3. We have added the description of the statistical analysis method for the in vivo assay to the legend for Figure 8:

“Statistical analysis: Two-tailed paired t-test. Data are presented as mean ± SEM. P-values ware for comparison with the tumors growing in untreated (UT) mice; ** p < 0.01; *** p < 0.001.”

Reviewer #3

1. The reviewer indicated “: In this revised manuscript, the authors have provided additional text in the introduction to explain the rationale for using an affibody-based PPEA polyplex in addition to their prior work that utilized an EGF-based complex. Figure quality and readability have been improved, and several typos and grammatical errors have been corrected.”

Since Reviewer #2 still had some issues with our figures, so we have made sure these will be clear and easy to read.

2. The reviewer states “. The biggest concern that remains is that the authors present in vitro data that the PPEA polyplex induces anti-tumor activity via an immune-mediated mechanism. However, they used an in vivo model that lacks an immune system, which demonstrated that PPEA polyplex induces anti-tumor activity via a non-immune mediated mechanism.” The reviewer adds “As the manuscript stands, their in vitro and in vivo data appear to be contradictory.”

Our in vitro and in vivo data are complementary – we observe killing in vitro without T-cell effects and we observe killing in vivo without T-cell effects. PolyIC is know to induce tumor cell killing once internalized, both directly and via cytokine activation of innate immunity via dendritic cells, NK cells, macrophages, eosinophils and neutrophils. In vitro the killing is essentially via the direct induction of apoptosis, although there maybe some indirect jilling through cytokines such as TNF-α. In vivo, there was direct killing, but the innate immune system of the nude mice is functional and also likely to kill of the tumor cells. There is no T-cell immune activation in the nude mice, so this means of extra targeting would not occur in our mouse model, but in syngeneic systems such as occurs in the clinical situation, not only would the direct and innate killing be induced, there should be T-cell tumor cytotoxicity as well. The ability of polyIC to kill human tumour cells in nude mice has been documented many times.

For example Talmadge, J. E., Adams, J., Phillips, H., Collins, M., Lenz, B., Schneider, M., Schlick, E., Ruffmann, R.,Wiltrout, R. H., and Chirigos, M. A.(1985). Immunomodulatory effects in mice of polyinosinic-polycytidylic acid complexed with poly-L-lysine and carboxymethylcellulose. Cancer Res. 45, 1058–1065

Kumar, H., Koyama, S., Ishii, K. J.,Kawai, T., and Akira, S. (2008). Cutting edge: cooperation of IPS-1- and TRIF-dependent pathways in polyIC-enhanced antibody production and cytotoxic T cell responses. J. Immunol. 180, 683–687.

Okada, H., Kalinski, P., Ueda, R., Hoji, A., Kohanbash, G., Donegan, T. E., Mintz, A. H., Engh, J. A., Bartlett, D. L., Brown, C. K., Zeh, H., Holtzman, M. P., Reinhart, T. A., Whiteside, T. L., Butterﬁeld, L. H., Hamilton, R. L., Potter, D. M., Pollack, I. F., Salazar, A. M., and Lieberman, F. S. (2011). Induction of CD8+ T-cell responses against novel glioma-associated antigen peptides and clinical activity by vaccinations with {alpha}-type 1 polarized dendritic cells and polyinosinic-polycytidylic acid stabilized by lysine and carboxymethyl cellulose

in patients with recurrent malignant glioma. J. Clin. Oncol. 29,330–336.

Vidit Singh, Anna Chernatynskaya, Lin Qi, Hsin-Yin Chuang, Tristan Cole, Vimalin Mani Jeyalatha, Lavanya Bhargava, W. Andrew Yeudall, Laszlo Farkas, and Hu Yang (2024)Liposomes-Encapsulating Double-Stranded Nucleic Acid (Poly I:C) for Head and Neck Cancer Treatment ACS Pharmacology & Translational Science 7 (5), 1612-1623DOI: 10.1021/acsptsci.4c00121

We acknowledge that the nude mouse model we used is not perfect, but it does show the ability of the PPEA-polyplex to reach the tumor, for the polyIC to be internalized and for the internalized polyIC to induce the killing of the tumor cells in vivo. These results indicate that the PPEA-polyplex is likely to be a potent anti-tumor agent in vivo and when an intact immune system is present the agent is likely to be even more effective. The suggested experiments with a syngeneic mouse tumor model would likely yield more information, but there are many difficulties and the syngeneic tumor model we tried led to ulceration in the mice. We could not justify attempting to reduce the side-effects. We believe that the readers will recognize the value of our results and the limitations of the nude mouse model.

3. The reviewer wrote “Additionally, while the authors present data that show differential response to PPEA polyplex based on EGFR expression, they only present data for EGFR surface expression for 3 of the cell lines used in the study, despite requests from two reviewers to provide data for additional cell lines.”

We had measured the surface EGFR levels for many more cell lines; figure 3B has been expanded to include the EGFR surface expression levels for 16 cell lines. We used a smaller number of cell lines for the targeting studies, but our cytotoxicity results were encouraging and along with our cytokine release studies and cell activation studies, we believe our results are a strong indication of the potential of this reagent for cancer therapy trials.

4. The reviewer states “Lastly, since their point-by-point response did not include the reviewer comments, I was unable to fully ascertain if all comments from reviewers 1 and 2 were addressed” From the responses of reviewers 1 and 2 – they were almost entirely satisfied with this revision – above we have responded to reviewer #2 two comments.

Specific comments:

1. Page 21 The reviewer is correct. We now only reference Supplementary Figure 5 once and we have improved the sentence structure.

2. Table 3 “In the results section of the text (page 22 line 12), the authors incorrectly state that these are pro-inflammatory cytokines.” Despite this comment, there is an extensive literature indicating that IP10 and Gro-α are pro-inflammatory cytokines. For example:

IP 10

Zhang Z, Kaptanoglu L, Tang Y, Ivancic D, Rao SM, Luster A, Barrett TA, Fryer J. IP-10-induced recruitment of CXCR3 host T cells is required for small bowel allograft rejection. Gastroenterology. 2004 Mar;126(3):809-18. doi: 10.1053/j.gastro.2003.12.014. PMID: 14988835

Dufour JH, Dziejman M, Liu MT, Leung JH, Lane TE, Luster AD (April 2002). "IFN-gamma-inducible protein 10 (IP-10; CXCL10)-deficient mice reveal a role for IP-10 in effector T cell generation and trafficking". Journal of Immunology. 168 (7): 3195-204. Bibcode:2002JImm..168.3195D. doi:10.4049/jimmunol.168.7.3195. PMID 1190 7072.

Gro-alpha

Ichikawa A, Kuba K, Morita M, Chida S, Tezuka H, Hara H, Sasaki T, Ohteki T, Ranieri VM, dos Santos CC, Kawaoka Y, Akira S, Luster AD, Lu B, Penninger JM, Uhlig S, Slutsky AS, Imai Y. CXCL10-CXCR3 enhances the development of neutrophil-mediated fulminant lung injury of viral and nonviral origin. Am J Respir Crit Care Med. 2013 Jan 1;187(1):65-77. doi: 10.1164/rccm.201203-0508OC. Epub 2012 Nov 9. PMID: 23144331; PMCID: PMC3927876.

Consequently, we have decided to retain the term pro-inflammatory cytokines for these molecules.

3. The reviewer states “Page 23 lines 22 and 29, Table 4. The authors also incorrectly use the term cytotoxic cytokines when referring to pro-inflammatory cytokines.” It is correct that these are pro-inflammatory cytokines, but they are also referred to as cytotoxic cytokines. Foe example:

TNF-alpha

Li M, Beg AA. Induction of necrotic-like cell death by tumor necrosis factor alpha and caspase inhibitors: novel mechanism for killing virus-infected cells. J Virol. 2000 Aug;74(16):7470-7. doi: 10.1128/jvi.74.16.7470-7477.2000. PMID: 10906200; PMCID: PMC112267.

Gregory T. Baxter, Richard C. Kuo, Orla J. Jupp, Peter Vandenabeele, David J. MacEwan (1999)

Tumor Necrosis Factor-α Mediates Both Apoptotic Cell Death and Cell Proliferation in a Human Hematopoietic Cell Line Dependent on Mitotic Activity and Receptor Subtype Expression Journal of Biological Chemistry, Volume 274, Issue 14, p9539-9547,

https://doi.org/10.1074/jbc.274.14.9539

Ksontini R, MacKay SLD, Moldawer LL. Revisiting the Role of Tumor Necrosis Factor α and the Response to Surgical Injury and Inflammation. Arch Surg. 1998;133(5):558–567. doi:10.1001/archsurg.133.5.558

In the context of our manuscript we believe that the description of IFNγ and TNFα as cytotoxic cytokines is accurate and we have decided to retain that nomenclature.

IFN-gamma

Badie B, Schartner J, Vorpahl J, Preston K. Interferon-gamma induces apoptosis and augments the expression of Fas and Fas ligand by microglia in vitro. Exp Neurol. 2000 Apr;162(2):290-6. doi: 10.1006/exnr.1999.7345. PMID: 10739635.

Fang C, Weng T, Hu S, Yuan Z, Xiong H, Huang B, Cai Y, Li L, Fu X. IFN-γ-induced ER stress impairs autophagy and triggers apoptosis in lung cancer cells. Oncoimmunology. 2021 Aug 10;10(1):1962591. doi: 10.1080/2162402X.2021.1962591. PMID: 34408924; PMCID: PMC8366549.

4. The reviewer states “The formatting of the tables at the end of the manuscript still makes them difficult to read.” In our first revision we did adjust the Tables to improve the formatting, but the conversion to the .pdf format was still not satisfactory. We have made further adjustments to make sure they format well in the .pdf.

5. The reviewer is correct - All of the inappropriate “Flow”s have now been changed to “flow”s

6. The reviewer is correct - All of the references to transfection, transfected and transfecting have now been replaced with treatment, treated and treating.

We are grateful for the opportunity to work with the editors and the reviewers to improve our manuscript. We believe we have responded to all the comments in an appropriate way and hope that our manuscript is now suitable for publication in PlosOne

Yours sincerely

Antony Burgess

---

## [Decision Letter · Decision Letter 2]

1 Apr 2026

PONE-D-25-53010R2EGFR-targeted affibody–polyIC polyplex kills EGFR-overexpressing cancer cells without activating the EGFRPLOS One

Dear Dr.  Burgess,

Thank you for submitting your manuscript to PLOS ONE. After careful consideration, we feel that it has merit but does not fully meet PLOS ONE’s publication criteria as it currently stands. Therefore, we invite you to submit a revised version of the manuscript that addresses the points raised during the review process.

We look forward to receiving your revised manuscript.

Kind regards,

Hamidreza Montazeri Aliabadi

Academic Editor

PLOS One

Journal Requirements:

Additional Editor Comments:

As you would see reviewers appreciated your efforts in addressing their comments and clarifying important information, which I truly appreciate also. There is only one comment from Reviewer 3 that requires your attention.

Reviewer's Responses to Questions

**Comments to the Author**

1. If the authors have adequately addressed your comments raised in a previous round of review and you feel that this manuscript is now acceptable for publication, you may indicate that here to bypass the “Comments to the Author” section, enter your conflict of interest statement in the “Confidential to Editor” section, and submit your "Accept" recommendation.

Reviewer #2: All comments have been addressed

Reviewer #3: All comments have been addressed

2. Is the manuscript technically sound, and do the data support the conclusions?

Reviewer #2: Yes

Reviewer #3: Yes

3. Has the statistical analysis been performed appropriately and rigorously? 

Reviewer #2: Yes

Reviewer #3: Yes

4. Have the authors made all data underlying the findings in their manuscript fully available?

Reviewer #2: Yes

Reviewer #3: Yes

5. Is the manuscript presented in an intelligible fashion and written in standard English?

Reviewer #2: Yes

Reviewer #3: Yes

6. Review Comments to the Author

Reviewer #2: (No Response)

Reviewer #3: In this second revision of the manuscript, the authors have provided additional editing to the figures and text to improve figure clarity and readability of the manuscript. They have also addressed the concern about using an in vivo model that lacks adaptive immunity, providing justification for using this model (presence of adaptive immunity) and a potential mechanism of action for the anti-tumor activity of PolyIC (apoptosis) that was lacking from the initial version of the manuscript and the first revision. Importantly, they have added EGFR surface expression data for additional cell lines in Figure 3. Additionally, the authors provide citations to justify their use of differential cytokine terminology in the text.

Overall, the largest concern regarding the choice of in vivo model has been sufficiently addressed in the author’s rebuttal to the reviewer comments. However, this discussion to justify their model and explain the mechanism of action of PolyIC in vivo in a model that lacks adaptive immunity was not included in the revised text. The current version of the manuscript discussion is not specific regarding mechanism of action of PolyIC and only refers to “intracellular killing pathways.” Once further justification of use of nude mice and possible mechanism of action of PolyIC has been added to the manuscript, I feel the manuscript will be suitable for publication.

7. PLOS authors have the option to publish the peer review history of their article (what does this mean?). If published, this will include your full peer review and any attached files.

Reviewer #2: No

Reviewer #3: No

You may also use PLOS’s free figure tool, NAAS, to help you prepare publication quality figures: https://journals.plos.org/plosone/s/figures#loc-tools-for-figure-preparation

---

## [Author Response · Author response to Decision Letter 3]

6 Apr 2026

Author Responses to Final comments by Reviewer 3

In response to the reviewer’s final concern regarding the use of the nude mouse model and the mechanism of action of polyIC in vivo, we have revised the Discussion to explicitly address these points. Specifically, we clarify that although nude mice lack T cells, they retain functional innate immune components, including NK cells, macrophages, dendritic cells and other effector cells, which can contribute to tumor cell killing.

We have also expanded the Discussion to describe the mechanism of action of polyIC following intracellular delivery, including direct tumor cell killing via dsRNA-induced apoptotic pathways and activation of innate immune responses through Toll-like receptor 3 (TLR3) and cytosolic sensors such as melanoma differentiation–associated protein 5 (MDA5).

Appropriate citations have been added to support these additions, in particular clarifying the mechanism by which polyIC mediates its anti-tumor activity in the absence of adaptive immunity.

---

## [Decision Letter · Decision Letter 3]

14 Apr 2026

EGFR-targeted affibody–polyIC polyplex kills EGFR-overexpressing cancer cells without activating the EGFR

PONE-D-25-53010R3

Dear Dr. Burgess,

We’re pleased to inform you that your manuscript has been judged scientifically suitable for publication and will be formally accepted for publication once it meets all outstanding technical requirements.

Kind regards,

Hamidreza Montazeri Aliabadi

Academic Editor

PLOS One

Additional Editor Comments (optional):

Reviewers' comments:

Reviewer's Responses to Questions

**Comments to the Author**

1. If the authors have adequately addressed your comments raised in a previous round of review and you feel that this manuscript is now acceptable for publication, you may indicate that here to bypass the “Comments to the Author” section, enter your conflict of interest statement in the “Confidential to Editor” section, and submit your "Accept" recommendation.

Reviewer #3: All comments have been addressed

2. Is the manuscript technically sound, and do the data support the conclusions?

Reviewer #3: Yes

3. Has the statistical analysis been performed appropriately and rigorously? 

Reviewer #3: Yes

4. Have the authors made all data underlying the findings in their manuscript fully available?

Reviewer #3: Yes

5. Is the manuscript presented in an intelligible fashion and written in standard English?

Reviewer #3: Yes

6. Review Comments to the Author

Reviewer #3: (No Response)

7. PLOS authors have the option to publish the peer review history of their article (what does this mean?). If published, this will include your full peer review and any attached files.

Reviewer #3: No

---

## [Editor Report · Acceptance letter]

PONE-D-25-53010R3

PLOS One

Dear Dr. Burgess,

I'm pleased to inform you that your manuscript has been deemed suitable for publication in PLOS One. Congratulations! Your manuscript is now being handed over to our production team.

Kind regards,

on behalf of

Dr. Hamidreza Montazeri Aliabadi

Academic Editor

PLOS One